# Urine-derived exosomes from individuals with IPF carry pro-fibrotic cargo

Sharon Elliot[1]*, Paola Catanuto[1], Simone Pereira-simon[1], Xiaomei Xia[2], Shahriar Shahzeidi[3], Evan Roberts[4], John Ludlow[5], Suzana Hamdan[6,7], Sylvia Daunert[6,7,8], Jennifer Parra[2], Rivka Stone[9], Irena Pastar[9], Marjana Tomic-Canic[9], Marilyn K Glassberg[1,2,10]

[1]DeWitt Daughtry Family Department of Surgery, University of Miami Leonard M. Miller School of Medicine, Miami, United States; [2]Department of Medicine, Division of Pulmonary, Critical Care and Sleep, University of Miami, Miami, United States; [3]Medical Director, Grand Health Institute, Miami, United States; [4]Cancer Modeling Shared Resource Sylvester Comprehensive Cancer Center, University of Miami, Miami, United States; [5]ZenBio Inc., Durham, United States; [6]Department of Biochemistry and Molecular Biology, University of Miami, Miller School of Medicine, Miami, United States; [7]Dr. JT Macdonald Foundation Biomedical Nanotechnology Institute, University of Miami Miller School of Medicine, Miami, United States; [8]Miami Clinical and Translational Science Institute, University of Miami Miller School of Medicine, Miami, United States; [9]Wound Healing and Regenerative Medicine Research Program, Dr Phillip Frost Department of Dermatology and Cutaneous Surgery, University of Miami, Miami, United States; [10]Department of Medicine, Stritch School of Medicine, Loyola University Chicago, Chicago, United States

*For correspondence:
selliot28@hotmail.com

## Abstract

**Background:** MicroRNAs (miRNA) and other components contained in extracellular vesicles may reflect the presence of a disease. Lung tissue, sputum, and sera of individuals with idiopathic pulmonary fibrosis (IPF) show alterations in miRNA expression. We designed this study to test whether urine and/or tissue derived exosomal miRNAs from individuals with IPF carry cargo that can promote fibrosis.

**Methods:** Exosomes were isolated from urine (U-IPFexo), lung tissue myofibroblasts (MF-IPFexo), serum from individuals with IPF (n=16) and age/sex-matched controls without lung disease (n=10). We analyzed microRNA expression of isolated exosomes and their in vivo bio-distribution. We investigated the effect on ex vivo skin wound healing and in in vivo mouse lung models.

**Results:** U-IPFexo or MF-IPFexo expressed *miR-let-7d, miR-29a-5p, miR-181b-3p and miR-199a-3p* consistent with previous reports of miRNA expression obtained from lung tissue/sera from patients with IPF. In vivo bio-distribution experiments detected bioluminescent exosomes in the lung of normal C57Bl6 mice within 5 min after intravenous infusion, followed by distribution to other organs irrespective of exosome source. Exosomes labeled with gold nanoparticles and imaged by transmission electron microscopy were visualized in alveolar epithelial type I and type II cells. Treatment of human and mouse lung punches obtained from control, non-fibrotic lungs with either U-IPFexo or MF-IPFexo produced a fibrotic phenotype. A fibrotic phenotype was also induced in a human ex vivo skin model and in in vivo lung models.

**Conclusions:** Our results provide evidence of a systemic feature of IPF whereby exosomes contain pro-fibrotic miRNAs when obtained from a fibrotic source and interfere with response to tissue injury as measured in skin and lung models.

**Funding:** This work was supported in part by Lester and Sue Smith Foundation and The Samrick Family Foundation and NIH grants R21 AG060338 (SE and MKG), U01 DK119085 (IP, RS, MTC).

### Editor's evaluation

This study presents valuable findings that exosomes support the development of idiopathic pulmonary fibrosis. The evidence is solid in both in vitro and in vivo bleomycin models of fibrosis. The authors conclude that exosomes might indicate that idiopathic pulmonary fibrosis is a systemic disease.

## Introduction

Over 200,000 older Americans currently suffer from chronic age-associated fibrotic lung disease (*Cordier and Cottin, 2013*; *Lederer and Martinez, 2018*; *Richeldi et al., 2017*) annually (*Hecker, 2018*; *Lederer and Martinez, 2018*; *Martinez et al., 2017*; *Raghu et al., 2016*). Alveolar epithelial cell damage, proliferation of fibroblasts and extracellular matrix (ECM) accumulation lead to irreversible disruption of the lung architecture (*Kinoshita and Goto, 2019*). While the etiology of IPF remains unknown, there is an ongoing need for understanding mechanisms of disease and diagnostic and prognostic biomarkers of disease (*Drakopanagiotakis et al., 2018*; *Tzouvelekis et al., 2016*).

Extracellular vesicles (EVs), including exosomes and microvesicles, are potential candidates for elucidation of a biomarker signature for IPF and other fibrotic diseases (*H Rashed et al., 2017*). As a key component of cell–cell communication, EVs deliver miRNAs, mRNA, and tRNA into target cells (*Yáñez-Mó et al., 2015*). In addition, their systemic distribution in body fluids may reflect the presence of disease. The ease of collection and presence of exosomes in most body fluids, including blood, breast milk, saliva, urine, bile, pancreatic juice, cerebrospinal, and peritoneal fluids, have facilitated recent studies investigating their use as diagnostic and prognostic markers of disease (*Keller et al., 2011*; *Raposo and Stoorvogel, 2013*). Because exosomes are released by all cells and contain cargo from the cell of origin, it is conceivable that they deliver pathogenic materials leading to a disease phenotype. Whether the content of exosomes from different sources are the same has not been studied previously.

Molecular analysis of lung biopsies from individuals with IPF reveal a unique miRNA transcriptome compared with the miRNA transcriptome found from non-fibrotic lung biopsy samples providing evidence supporting the disease conferring properties of exosomes (*Guiot et al., 2019*). In serum isolated from individuals with IPF compared to control serum samples, there were 47 differentially expressed miRNAs (*Yang et al., 2015*). Njock et al. reported a signature of *miR-142–3 p, miR-33a-5p, miR-let-7d-5p* in sputum of individuals with IPF (*Njock et al., 2019*). Parimon et al. found that in lung tissue EVs isolated from mouse alveolar spaces and IPF lungs, TGF-β and Wnt signaling were increased after bleomycin (Bleo) treatment. These data trigger a hypothesis that dysregulation of miRNAs packaged in exosomes in diseased sources are involved in the development and progression of fibrosis (*Guiot et al., 2019*; *Parimon et al., 2019*).

To the best of our knowledge, there are no published studies using exosomes from urine of individuals with IPF (U-IPFexo) or any urine-derived exosomes in the investigation of fibrotic lung disease. In the present study we isolated and characterized exosomes isolated from the urine of 16 male individuals with IPF and 10 age and sex-matched control individuals to test the hypothesis that U-IPFexo cargo could promote a fibrotic phenotype. We found a a fibrotic miRNA expression pattern in serum and urine derved from the same individual with IPF. Our results also confirmed that several miRNAs previously reported in the lung, serum, and sputum from individuals with IPF (*Njock et al., 2019*; *Pandit and Milosevic, 2015*; *Rajasekaran et al., 2015*) could be identified in urine, lung, and serum IPFexo.

Intravenous infusion of urine exosomes isolated from both normal and diseased individuals using in vivo bioluminescent imaging system, revealed fairly rapid biodistribution to the lung. Ex vivo human and mouse lung punch studies and in vivo mouse models were used to functionally assess the effect of tissue (lungs) and systemic (urine and serum) exosomes isolated from individuals with IPF and 'control' exosomes isolated from urine and fibroblasts from control lungs (*Habermann et al., 2019*). U-IPFexo conveyed a pro-fibrotic lung phenotype and inhibited skin tissue repair. The unique signature was

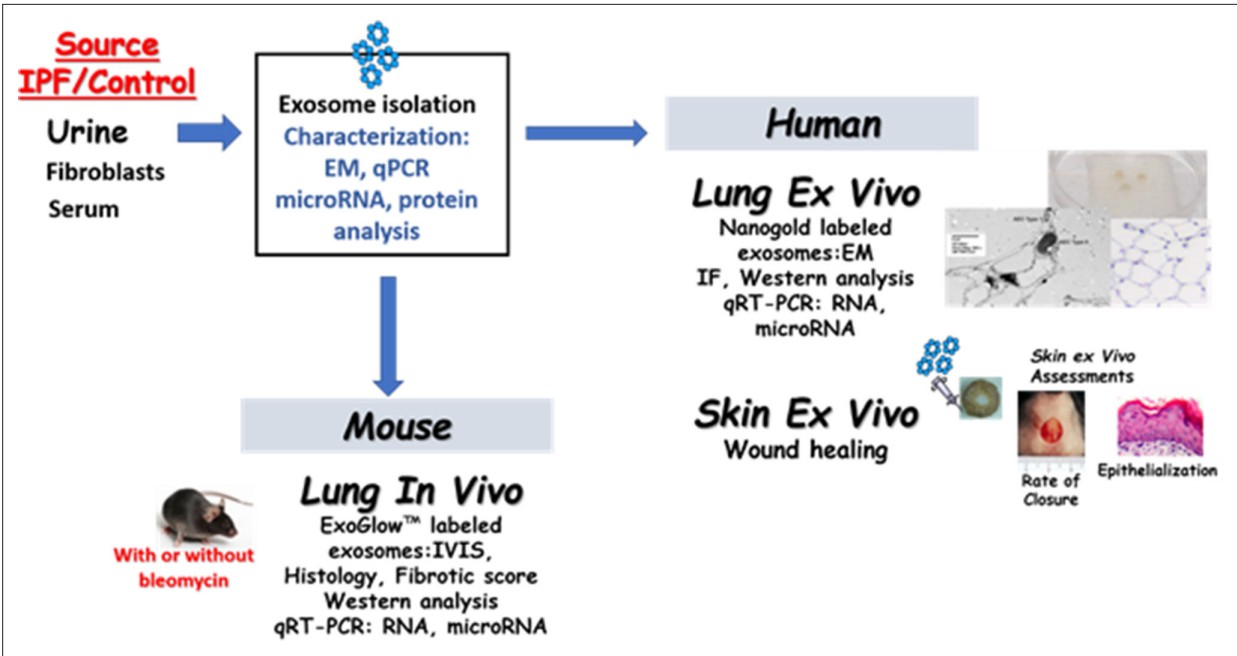

**Figure 1.** Overview of experimental details and design.

specific to IPFexo, and not only transferred a pro-fibrotic lung phenotype to control mice but further increased markers of lung fibrosis in vivo. Since body weight is independently associated with survival of individuals with IPF, we used body weight as a surrogate for stage of disease and found that mice lost more body weight after treatment with U-IPFexo than with Bleo treatment alone (*Mogulkoc et al., 2018*; *Nakatsuka et al., 2018*). We also included urine exosomes from subjects with non-cystic fibrosis (non-CF) bronchiectasis or asthma (non-fibrotic lung diseases) in in vivo experiments. Mice receiving these exosomes did not exhibit changes in fibrotic markers. These findings suggest that markers of lung fibrosis are specific, may be detected systemically, outside of the lung, and can communicate the pathology to different tissues. Importantly, urine exosomes have the potential to be used as disease-defining biomarkers in patients with IPF.

## Materials and methods

A complete list of source data including antibodies and primer sequences are included as supplement 1. An overview of the study is shown in *Figure 1*.

### Urine and blood collection

Random urine samples were collected (between 10am-2pm) from male control individuals (ages 55–77) or individuals seen in the Interstitial Lung Disease (ILD, subjects with IPF ages 55–79; subjects with non-CF bronchiectasis ages 73–86; and subjects with asthma ages 36–61) clinic at the University of Miami Hospital. Written informed consent was obtained from the participants with an approved Institutional Review Board protocol (IRB# 20060249) at the University of Miami Miller School of Medicine. Control urines were obtained from subjects that did not have underlying kidney, heart, or lung comorbidities and had normal albumin creatinine ratios sampled in an aliquot of each urine. Urine was processed within 3 hr of collection and spun at 3000xg for 15 min to remove sediment; supernatant was aliquoted at 10 ml/tube. Patients with known kidney disease or diabetes were excluded from the study and a small aliquot of urine was used to measure urine albumin and creatinine to exclude samples with albuminuria (Creatinine LiquiColor Test, Stanbio Laboratory Boerne, TX and albumin Elisa, Bethyl laboratories, Montgomery, TX). Tubes were frozen at –80 ° C until exosome isolation. Blood was collected during the same clinic visit as urine collection. Blood was left to clot for up to an hour and serum separated after centrifugation at ~2500 RPM for 15 min. Serum was aliquoted and frozen at –80 ° C until exosome isolation. Samples were selected for inclusion in the present study

**Table 1.** Male IPF (A group), non-CF bronchiectasis (B group), or asthma (C group) urine-derived exosomes (Age of subject at collection).

| Subject number | Age of subject at collection | Ethnicity | FEV1 (liters) | FVC (liters) | FEV/FVC (%) | FEV/FVC (predicted) | DLCO (% reference) |
|---|---|---|---|---|---|---|---|
| A4 | 72 | Caucasian | 2.93 | 3.75 | 78 | 77 | 11.8 (47) |
| A6 | 79 | Caucasian | 2.36 | 2.91 | 81 | 67 | 16.2 (101) |
| A26 | 69 | Hispanic | 1.67 | 1.98 | 84 | 72 | NT |
| A35 | 69 | Hispanic | 1.71 | 2.22 | 70 | 73 | NT |
| A37 | 69 | Hispanic | 2.7 | 3.21 | 84 | 77 | 10.5 (44) |
| A62 | 67 | Hispanic | 1.94 | 2.33 | 82 | 77 | 8.7 (33) |
| A74 | 75 | Hispanic | 2.48 | 2.77 | 80 | 74 | 9.9 (49) |
| A77 | 55 | Hispanic | 1.14 | 1.27 | 90 | 78 | 6.7 (32) |
| A80 | 70 | Caucasian | 2.09 | 2.62 | 78.9 | 87.49 | 11.8 (48) |
| A83 | 68 | Hispanic | 2.39 | 2.69 | 86 | 75 | 10.5 (49) |
| A84 | 67 | Caucasian | 1.78 | 2.12 | 84 | 84 | 14.1 (57) |
| A88 | 66 | Hispanic | 1.36 | 1.39 | 98 | 75 | 2.6 (11) |
| A90 | 67 | Caucasian | 2.48 | 2.85 | 88.9 | 74 | 12.8 (45) |
| A103 | 72 | Hispanic | 1.95 | 2.64 | 74 | 75 | 13.4 (59) |
| A104 | 76 | Caucasian | 2.95 | 3.27 | 90 | 65 | 14.6 (58) |
| A105 | 62 | Hispanic | 1.47 | 1.69 | 87 | 78 | 11.7 (41) |
| B1 | 73 | Caucasian | 1.04 | 2.06 | 50 | 79.91 | 9.6 (48) |
| B10 | 70 | Hispanic | 1.33 | 2.89 | 46 | 74 | 21.29 (69.5) |
| B13 | 86 | Hispanic | 2.54 | 3.59 | 70.7 | 74 | 23.08 (76.78) |
| C1 | 36 | Hispanic | 3.67 | 4.75 | 77.16 | 82.89 | 31.49 (92) |
| C4 | 61 | Caucasian | 3.38 | 4.72 | 71.65 | 79.09 | 27.77 (96) |
| C7 | 44 | Caucasian | 3.08 | 4.57 | 67.3 | 70 | 27.83 (96) |

NT, not tested.

after review of imaging confirming a definitive UIP pattern (*Raghu et al., 2022*). Lung function (forced vital capacity [FVC], $FEV_1$, $FEV_1$: FVC%) and diffusion capacity (DLCO) performed at several facilities, were obtained from subjects' medical records closest to the timing of the sample (*Table 1*).

## Cell culture

Myofibroblasts isolated from male individuals with IPF and fibroblasts isolated from male control lungs were propagated and characterized as previously described (*Elliot et al., 2019*). All cell lines tested negative for mycoplasma. Two IPF and one control cell isolate was purchased from Lonza (catalog number CC7231 and CC2512, MSCBM, Lonza Inc, Walkersville, MD). Cells (passages 2 or 3) were grown until 80% confluence in 30 T175 flasks in Lonza media. At the time of collection, media was aspirated, and each flask washed three times with PBS to remove serum and serum proteins. After serum-free medium was added back to each flask for an additional 48 hr, the supernatant was collected and exosomes isolated and characterized (See Zen-Bio Inc, Durham, NC method below).

## Exosome Isolation (conditioned tissue culture medium, urine), characterization, and RNA isolation

A 4 °C sample was centrifuged at 3000 x g for 20 min at room temperature in a swinging bucket rotor to remove large cells and debris. The clarified supernatant was collected and ultracentrifuged at 100,000 x g for 2 hr, in a fixed angle rotor at 4 °C, to pellet exosomes. The exosome pellet was resuspended in a minimum volume of DPBS (approximately 120 µL/ultracentrifugation tube). Exosomes were characterized using a Thermo NanoDrop spectrophotometer for protein determination and approximate RNA concentration by direct absorbance; exosomes were not lysed, stained, or RNA extracted prior to taking these measurements. Particle diameter and concentration was assessed by tunable resistive pulse sensing TRPS; (qNano, Izon Science Ltd Christchurch, New Zealand), using a NP150 nanopore membrane at a 47 mm stretch. The concentration of particles was standardized using multi-pressure calibration with carboxylated polystyrene beads of a defined size (nm diameter) and at a defined concentration (particles/mL). The samples were deidentified so that the experimenter performing the isolation was blinded. MACSPlex surface protein analysis (Miltenyl Biotech, Auburn, CA) was performed according to manufacturer's directions on two samples of urine derived exosomes.

For RNA isolation the Norgen Preserved Blood RNA Purification Kit I (Norgen Biotek Corp, Ontario, Canada) was used according to manufacturer's directions. Briefly, for every 100 µl of exosome preparation, 300 µL of lysis solution was added. After a short vortex, 400 µL of 95–100% ethanol was added. 600 µl of the lysate containing ethanol was loaded onto the column and centrifuged for 1 min at ≥3500 x g (~6000 RPM). The column was washed three times with wash solution and centrifuged each time for 1 min (14,000 x g). The column was spun for an additional 2 min in order to thoroughly dry the resin and the column contents eluded with 50 µl of Elution Solution by centrifuging for 2 min at 200 x g (~2000 RPM), followed by 1 min at 14,000 x g.

## Biodistribution

Mice were fed an alfalfa-free diet to prevent auto fluorescence of tissues (Perkin-Elmer personal communication). Exosomes were labeled with ExoGlow-Vivo EV Labeling Kit (System Biosciences, Palo Alto, CA). Labeled exosomes or PBS containing exosomes were injected via tail vein into 17- to 18-month-old C57BL/6 mice. Mice were anesthetized with isoflurane and imaged using an IVIS Spectrum In Vivo Imaging System (Perkin Elmer). The mice were imaged at 5, 10, 15, 30, 60, 90 min, 2, 4, 6, and 24 hr as well as at 2, 3, and 19 days post injection. Imaging data was analyzed using Living Image Software (Perkin Elmer). Control experiments were conducted with dye alone.

## Transmission electron microscopy

For conventional transmission electron microscopy (TEM), exosome pellets were placed in a droplet of 2.5% glutaraldehyde in PBS buffer at pH 7.2 and fixed overnight at 4 °C. The samples were rinsed in PBS (3 times, 10 min each) and post-fixed in 1% osmium tetroxide for 60 min at room temperature. The samples were then embedded in 10% gelatin, fixed in glutaraldehyde at 4 °C and cut into several blocks (smaller than 1 mm$^3$). The samples were dehydrated for 10 min per step in increasing concentrations of alcohol (30%, 50%, 70%, 90%, 95%, and 100%×3). Next, pure alcohol was replaced with propylene oxide, and the specimens were infiltrated with increasing concentrations (25%, 50%, 75%, and 100%) of Quetol-812 epoxy resin mixed with propylene oxide for a minimum of 3 hr per step. The samples were embedded in pure, fresh Quetol-812 epoxy resin and polymerized at 35 °C for 12 hr, 45 °C for 12 hr, and 60 °C for 24 hr. Ultrathin sections (100 nm) were cut using a Leica UC6 ultramicrotome and post-stained, first with uranyl acetate for 10 min and then with lead citrate for 5 min at room temperature, prior to observation using the FEI Tecnai T20 transmission electron microscope (Hillsboro, TX, USA) operated at 120 kV.

## Gold nanoparticle labelling of exosomes

Gold nanoparticles (nanospheres) modified with branched polyethylenimine (BPEI) of 10 nm size were used (*nanoComposix, co*). The original concentration of gold nanoparticles purchased was 1 mg/mL (5.7E+13 particles/mL). We preformed experiments to optimize the ratio between the exosome particles and the modified gold nanoparticles with the final prep containing 10*$^8$ exosomes: 0.001 mg of the modified gold nanoparticles. Briefly, gold nanosuspensions were sonicated prior to the experiment. The concentration of gold nanoparticles was adjusted based on the concentration of each

exosome prep and the desired volume for injection. The mixture was vortexed and then placed in a thermomixer (*Eppendorf ThermoMixer F1.5*) at 37 °C and spun at 300 rpm. After 3 hr, the mixture was vortexed, allowed to sit at room temperature for 15 min, and placed at 4 °C until the next day.

## Ex vivo lung punches

Cannulated lungs from 17- to 18-month-old male C57BL/6 mice were filled with warm low melting point agarose (3%, Sigma-Aldrich, St. Louis, MO). Lung segments were cooled on ice for 30 min to allow solidification of the agarose, punched to a diameter of 4 mm using a biopsy punch (Acuderm Inc Ft Lauderdale, FL) and transferred to an air-liquid interface with phenol red free minimal essential media (MEM) without serum. Human urine-derived or cell derived exosomes were injected into the lung punch with 29 G needles in a volume of 100 μl. The 100 μl volume of injection was divided equally into four injection sites on the punch. Punches were incubated at 37 °C in a humidified atmosphere of 5% $CO_2$ for 4 days. Parallel punches (injected and naïve) from the same experiment were prepared for histology, immunofluorescence staining, RNA and protein purification. The normal histology of the punch was preserved after injection with PBS at four sites as shown by light and electron microscopic examination after PBS injection (*Figure 5—figure supplement 1*). For human lung, we injected a bronchial branch with 2% agarose and performed punch experiments as described above. Selected punches were injected with nanoparticle containing exosomes for EM experiments and collected within 24 hr of injection (see section above for nanoparticle labelling method). A series of punches were collected that did not receive any injections or were injected with nanoparticles alone. Four days after injection punches were fixed in 10% formalin (Sigma–Aldrich), processed for paraffin embedding and stained with trichrome to assess collagen content.

## Punch TEM

Individual punches were fixed in 2% glutaraldehyde in 0.05 M phosphate buffer and 100 mM sucrose, post-fixed overnight in 1% osmium tetroxide in 0.1 M phosphate buffer, dehydrated through a series of cold graded ethanol, and embedded in a mixture of EM-bed/Araldite (Electron Microscopy Sciences). One-μm-thick sections were stained with Richardson's stain for observation under a light microscope. A total of 100 nM sections were cut on a Leica UC7 ultramicrotome and stained with uranyl acetate and lead citrate. The grids were viewed t 80 kV in a JEOL JEM-1400 transmission electron microscope and images captured by an AMT BioSprint 12 digital camera.

## Immunofluorescence staining of lung punches

Formalin fixed paraffin embedded punches were processed as previously described for tissue (*Tashiro et al., 2015*). Fluorescent staining was performed using α-SMA (Abcam, Cambridge, MA) or anti-prosurfactant protein C (SPC, Abcam) and DAPI containing mounting medium (Vector Shield, Burlingame Ca).

## Real-time PCR

Total RNA was extracted from lung tissue homogenates. Amplification and measurement of target RNA was performed on the Step 1 real time PCR system as previously described (*Karl et al., 2006*). $α_v$-integrin, collagen type I and TGF-β were measured using RNA extracted from lung punches or lung tissue. ER subtypes, TGF-β and IGF-1 mRNA expression was measured using RNA extracted from exosomes. The TaqMan rRNA control reagents kit (Life Technologies) was used to detect *18* S rRNA gene, an endogenous control, and samples were normalized to the *18* S transcript content as previously described (*Potier et al., 2002*). For miRNA analyses, cDNA was generated using qScript microDNA cDNA Synthesis Kit (Quanta Biosciences, Beverly, MA) according to manufacturer's instructions (*Elliot et al., 2019*). Amplification of all miRNAs was performed on the QuantStudio 3 96well 0.2 ml Block Real-Time PCR System using specific primers, *miR-let-7d, miR-29a-5p, miR-34a-5p, miR-142–3 p, miR-199a-3p*, and *miR-181b* (IDT, Coralville, IA) using Real-Time SYBR Green qRT-PCR Amplification kit (Quanta Biosciences, Beverly, MA). U6 expression was used as a control for miRNA analyses, and relative expression was calculated using the comparative C(T) method (*Schmittgen and Livak, 2008*).

## Western blot

Punches or lung pieces were homogenized, and western analysis performed as previously described (*Glassberg et al., 2014*) using the Invitrogen mini-cell gel surelock cell module Xcell II vertical surelock

box (Thermofisher, Waltham, MA). For pAKT, AKT, c-Jun, Caveolin-1, ERα and β-actin, 5–25 μg of protein lysate was fractionated on 10% polyacrylamide gels. Immunoreactive bands were determined by exposing nitrocellulose blots to a chemiluminescence solution (Denville Scientific Inc, Metuchen, NJ) followed by exposure to Amersham Hyperfilm ECL (GE Healthcare Limited, Buckinghamshire, UK). To determine the relative amounts of protein densitometry Image J version 1.48 v (National Institutes of Health, Bethesda, MD) was utilized. All values from western blots were standardized to the corresponding β-actin band prior to comparative analyses.

## MMP activity

MMP activity was assessed in lung punches and lung tissue using a previously described method (*Glassberg et al., 2014*). Briefly, Novex 10% zymogram gels (Life Technologies) were incubated for 24 hr in a gelatinase solution, which allows the determination of total proteolytic MMP activities without interference from associated tissue inhibitors. Relative MMP activity was determined by densitometry using Image J (**NIH**).

## Ex vivo human skin wound model to evaluate functional effects of urine-derived exosomes

Human skin samples were obtained from healthy subjects following panniculectomy (abdominal skin; median age 44 years old). Informed consent was obtained per the requirements of the Institutional Review Board at the University of Miami (IRB protocol # 20070922). Under sterile conditions, subcutaneous fat was trimmed from skin prior to generating wounds. A 3 mm punch (Acuderm) was used to make wounds in the epidermis through the reticular dermis and 3 mm discs of epidermis were excised. Skin discs (8 mm), with the 3 mm epidermal wound in the middle, were excised using a 6 mm biopsy punch (Acuderm). Wounded skin specimens were immediately transferred to air-liquid interface with DMEM medium (BioWhittaker) supplemented with antibiotics-antimycotics and 10% fetal bovine serum (Gemimi Bio – Products). The skin samples were incubated at 37 °C in a humidified atmosphere of 5% $CO_2$ for 4 days. Tissues were fixed in 10% formalin (Sigma–Aldrich), processed for paraffin embedding and stained with hematoxylin and eosin to follow the rate of healing.

## Animal model

Seventen- to 18-month-old male C57BL/6 mice obtained from Jackson Laboratories were housed under specific pathogen-free conditions with food and water ad libitum. All experiments and procedures were approved by the Institutional Animal Care and Use Committee at the Leonard M. Miller School of Medicine at the University of Miami (Miami, FL), a facility accredited by the American Association for the Accreditation of Laboratory Animal Care. Following treatments all mice were housed one/per cage until sacrifice. Sample size was based on our published data (*Tashiro et al., 2015*). Starting with 8–10 mice per group we have >90% statistical power to detect a Cohen's effect size d>1.75 standard deviations.

## Bleomycin administration

After the administration of anesthesia, Bleomycin sulfate (Sigma-Aldrich) dissolved in 50 μl sterile saline at 2.5 U per kg of bodyweight was administered by direct intratracheal instillation via intubation. Control mice received 50 μl of sterile saline using the same method. Mice were weighed and sacrificed at 21 days post-Bleo administration.

## Exosome injections and time course

Urine-derived or cell-derived exosomes were thawed in a 37 ° C water bath and washed in PBS to remove the cell freezing solution immediately prior to injection. Twenty-four hours following Bleo administration, each animal received 100 μl either PBS (control) or 40 μg of exosomes in 100 μl of PBS by tail vein injection over a one minute period (*Tashiro et al., 2015*). This amount was calculated based on the number of exosomes derived from $10^5$ cells (number of cells utilized in whole cell experiments) (*Tashiro et al., 2015*). Some mice received Bleo +vehicle injection. Treatments were assigned by simple randomization and the technician was blinded to the treatments. Dose response experiments were conducted with 20 and 40 μg of exosomes. A separate set of naïve mice (not treated with Bleo), received the same exosome injections and were sacrificed at 21 days post injection.

## Murine lung tissue analysis/ immunohistochemistry

Left lung lobes were harvested for protein, zymography, and mRNA analysis (see methods above). For morphometry and histology studies, right lung lobes were inflated with 10% neutral buffered formalin (NBF) under 25 cm $H_2O$ pressure. The lungs were post fixed by immersion in 10% NBF for 24 hr and then transferred to PBS at 4 °C. Samples were paraffin-embedded and 4 µm sections were obtained for hematoxylin-eosin and Masson's Trichrome staining.

## Ashcroft scoring

Pulmonary fibrosis was assessed by a pulmonary pathologist blinded to the experimental groups using the semi-quantitative Ashcroft method (*Ashcroft et al., 1988*) on Masson's Trichrome-stained slides at ×20 magnification. Individual fields were assessed by systematically moving over a 32-square grid; each field was assessed for fibrosis severity and assigned a score on a scale of 0 (normal lung) to 8 (total fibrosis of the field) and an average was obtained for each slide.

**Table 2.** Male control (D group) urine-derived exosomes (Age of subject at collection).
No evidence of documented lung disease or abnormal PFTs.

| Subject number | Age of subject at collection | Ethnicity |
|---|---|---|
| D8 | 66 | Caucasian |
| D9 | 70 | Caucasian |
| D12 | 77 | Caucasian |
| D28 | 77 | Caucasian |
| D31 | 55 | Caucasian |
| D32 | 73 | Caucasian |
| D38 | 72 | Caucasian |
| D41 | 65 | Caucasian |
| D50 | 75 | Caucasian |
| D101 | 57 | Caucasian |

## Collagen content assessment by hydroxyproline content

Hydroxyproline content was determined according to the manufacturer's instructions (Hydroxyproline Assay Kit, Sigma-Aldrich, St. Louis, MO). Briefly, 2 mg lung fragments were weighed and homogenized in 100 µl of distilled water. An equal volume of 10 N HCl was added to the samples before drying at 49 ° C for 3 hr. A total of 50 µl of sample was loaded in the plate and incubated overnight at 37 ° C. A hydroxyproline standard curve was prepared according to a standard solution (0–1 µg/well). Hydroxyproline content was read at 557 nm, using the SoftMax Pro Software (Molecular Devices Corp, Sunnyvale, CA).

## Statistical analysis

Mean and SEM were determined using GraphPad Prism 9.0 (GraphPad Software, San Diego, CA). Statistically significant differences between groups were determined by using Kruskal-Wallis test and Mann-Whitney test for single comparisons. Given limited sample sizes in some experiments, data were determined to be normally distributed using Kolmogorov-Smirnov test and tested by one-way analysis of variance (ANOVA) and Tukey multiple comparison. Results were considered statistically significant at p<0.05. ANOVA was also used to analyse rate of epithelialization among treatment groups; p<0.05 was considered significant.

# Results

## Exosome isolation and characterization

Urine samples were collected from individuals diagnosed with IPF (A group) or non-CF bronchiectasis (B group) or asthma (C group, *Table 1*) and normal age and sex-matched controls (D group, *Table 2*, IRB # 20060249). Control urines were obtained from subjects that did not have underlying kidney, heart, or lung comorbidities and had normal albumin creatinine ratios

**Table 3.** Myofibroblast and control fibroblast-derived exosomes (Age of subject at collection).

| Male myofibroblasts IPF (Age of subject at collection) | Male fibroblast Control (Age of subject at collection) |
|---|---|
| 1 (52) | 5 (70) |
| 2 (83) | 6 (69) |
| 3 (73) | 7 (67) |
| 4 (74) | |

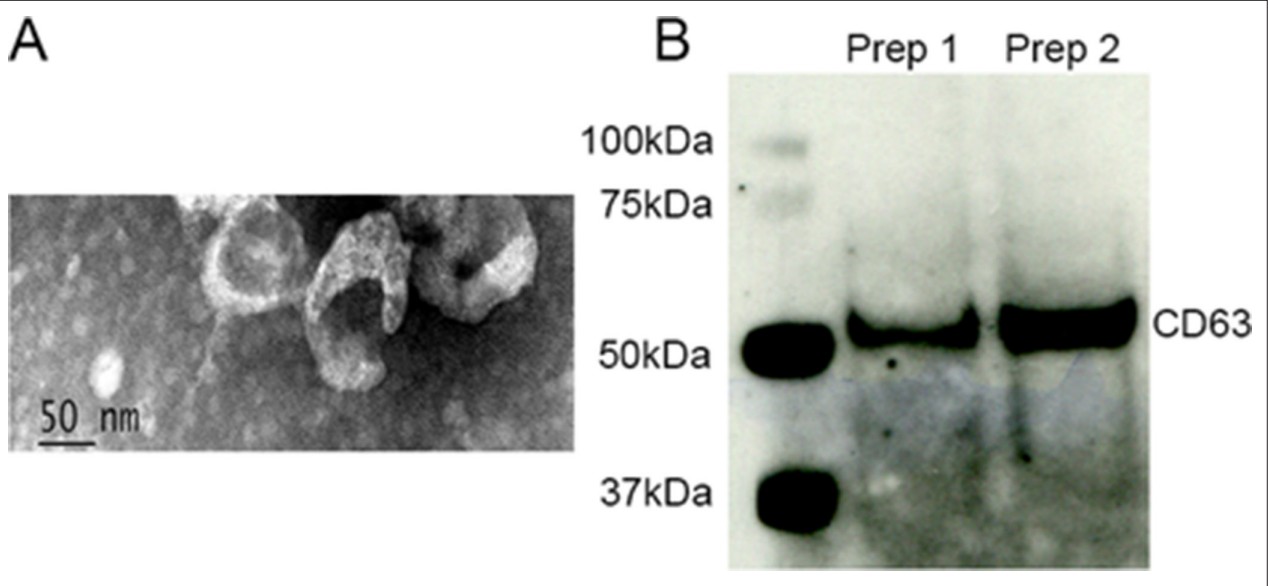

**Figure 2.** A Transmission electron microscopy of isolated exosomes. Image magnification Scale bar = 50 nm. 1B. Isolated exosomes express CD63.

sampled in an aliquot of each urine. IPF and control exosomes were isolated from myofibroblasts and fibroblasts (*Table 3*). Human urine, blood and cell derived exosome isolation were performed by Zen-Bio Inc (Durham, NC). Exosome size from urine, blood and cells was approximately 30–150 μm. EM performed on isolated exosomes (*Figure 2*) confirmed a size and shape previously reported for exosomes (*Gurunathan et al., 2019*). The presence of exosome markers CD1, CDC9, and CD63 were determined by MACSPlex analysis and CD63 (*Figure 2B*) was confirmed by western analysis. Other exosome markers present were CD9 and CD81.

## Determination of miRNAs expressed in urine- and serum-derived exosomes

To our knowledge, determination of the cargo of U-IPFexo has not been previously reported. U-IP-Fexo relative expression analysis revealed dysregulated expression of *miR-let-7D, miR-199, miR-29, and miR-181* (*Figure 3A, B, C and D*, *Figure 3—source data 1*) compared to urine-derived control exosomes. We compared serum-derived exosome miRNA to urine-derived exosomal miRNA from the same patient and found no difference in relative expression of selected miRNAs (*Figure 3E*, *Figure 3—source data 1*). These selected miRNAs have also been shown to be altered in lung tissue and serum from individuals with IPF (*Casanova et al., 2021*; *Omote and Sauler, 2020*; *Ortiz-Quintero et al., 2020*). These data suggest that cargo carried in urine exosomes from individuals with IPF may parallel cargo of serum-derived exosomes in the same individual.

## Detection of expression of mRNA from exosomes

Isolated exosomes were subjected to PCR to detect potential mRNA expression that are reflective of fibrotic pathways. As expected in pure exosome preparations (*Batagov and Kurochkin, 2013*), we did not detect mRNA expression of TGFβ, IGF or ER subtypes from normal or diseased exosomes.

## Tracking exosomes reveals a rapid systemic distribution

We assessed exosome signal and time course in 17- to 18-month-old male C57BL/6 mice using an in vivo bioluminescent imaging system. Regardless of source (U-IPFexo or control exosomes), exosomes were located in the lung within 5 min post-tail vein injection. Images were taken at 5 min, 15 min 30 min, 60 min, 90 min and 2, 4, 6, 24, and 48 hr (*Figure 4*) in all preparations. No adverse events were noted in mice followed for 20 days. All mice injected with U-IPFexo had a higher lung signal over time compared to mice injected with urine-derived age and sex-matched controls (*Figure 4—figure supplement 1A* ), suggesting that these exosomes may be retained longer in the lung than control exosomes. A set of mice sacrificed at 48 hr to quantify signals in heart/lung, spleen, liver, and kidney

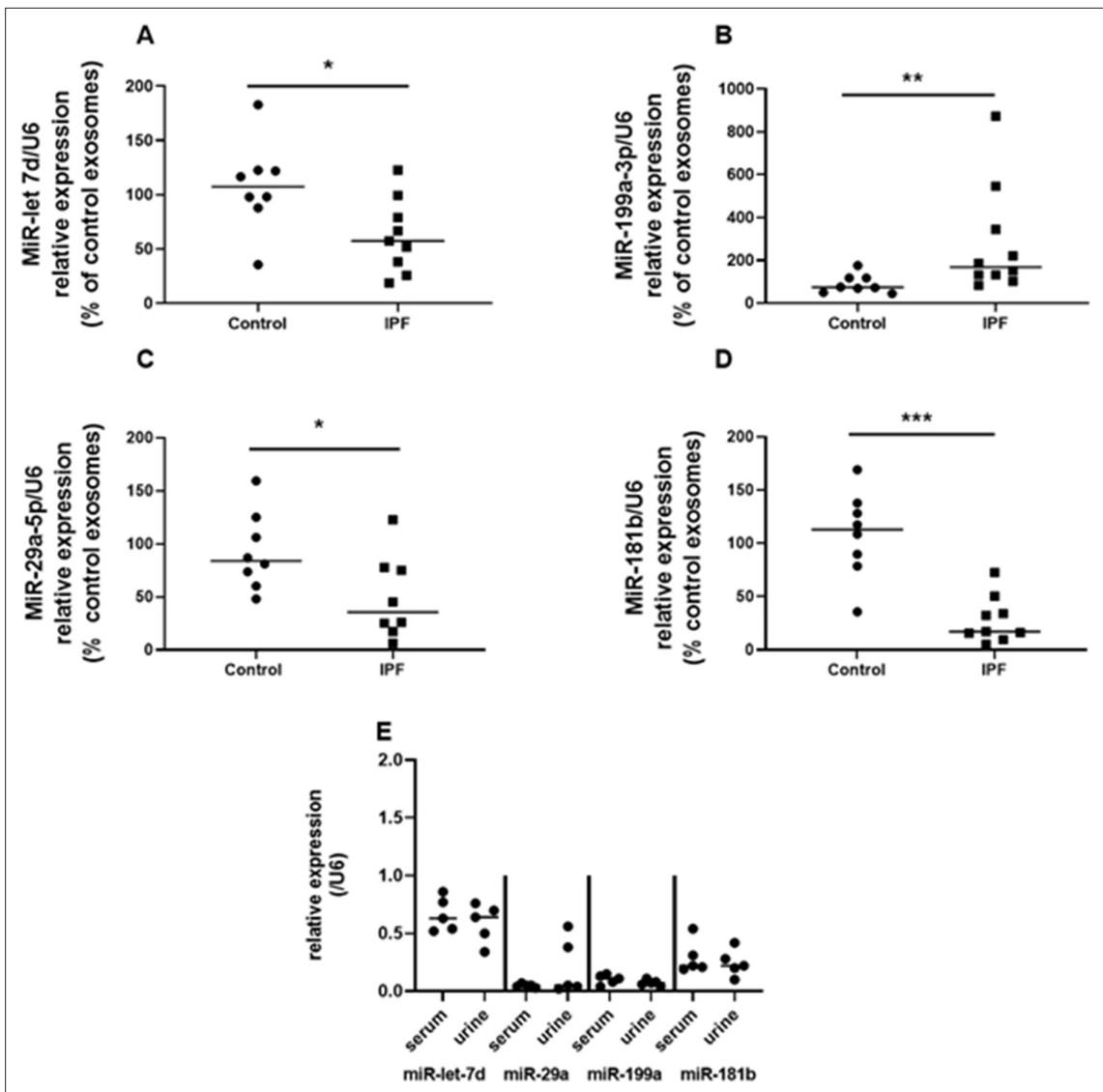

**Figure 3.** Expression of *miR-let-7d* (**A**), *miR-29a-5p* (**B**), *miR 181b-3p* (**C**) and *miR-199a-3p* (**D**) in urine-derived exosomes reveals a pattern corresponding to that reported in serum and whole lung of individuals with IPF. PCR was performed on extracted urine-derived exosomes as described in methods. Data are graphed as relative miRNA expression normalized to U6 and percent of control expression. * p<0.05, **p<0.01, *** p<0.001 compared to control exosomes. Each point represents an individual patient exosome sample. n=5–14 individual samples/group, *P* values were calculated by Mann-Whitney U test. E. Urine and serum-derived exosomes isolated from the same individuals with IPF have similar miRNA expression. Exosome isolation, RNA preparation and PCR performed as described in methods. n=5 individual samples of urine and serum-derived exosomes. Paired *T* test analysis was performed. *Figure 3—source data 1*.

The online version of this article includes the following source data for figure 3:

**Source data 1.** Raw data for *Figure 3*.

(*Figure 4—figure supplement 1C*) showed that the lung signal evident at 30 min (*Figure 4—figure supplement 1B*) was not evident at 48 hr as seen by IVIS suggesting clearance of the exosomes. Mice injected with PBS had no signal (*Figure 4C*).

## Transmission electron microscopy reveals uptake of exosomes into alveolar epithelial cells (AEC)

Exosomes were labeled with gold nanoparticles at varying ratios and processed for EM to determine the optimum ratio of nanoparticles to exosomes. Utilizing TEM, exosomes labeled with nanoparticles were visualized in mouse lung punches (*Figure 5—figure supplement 1A*) injected with urine-derived

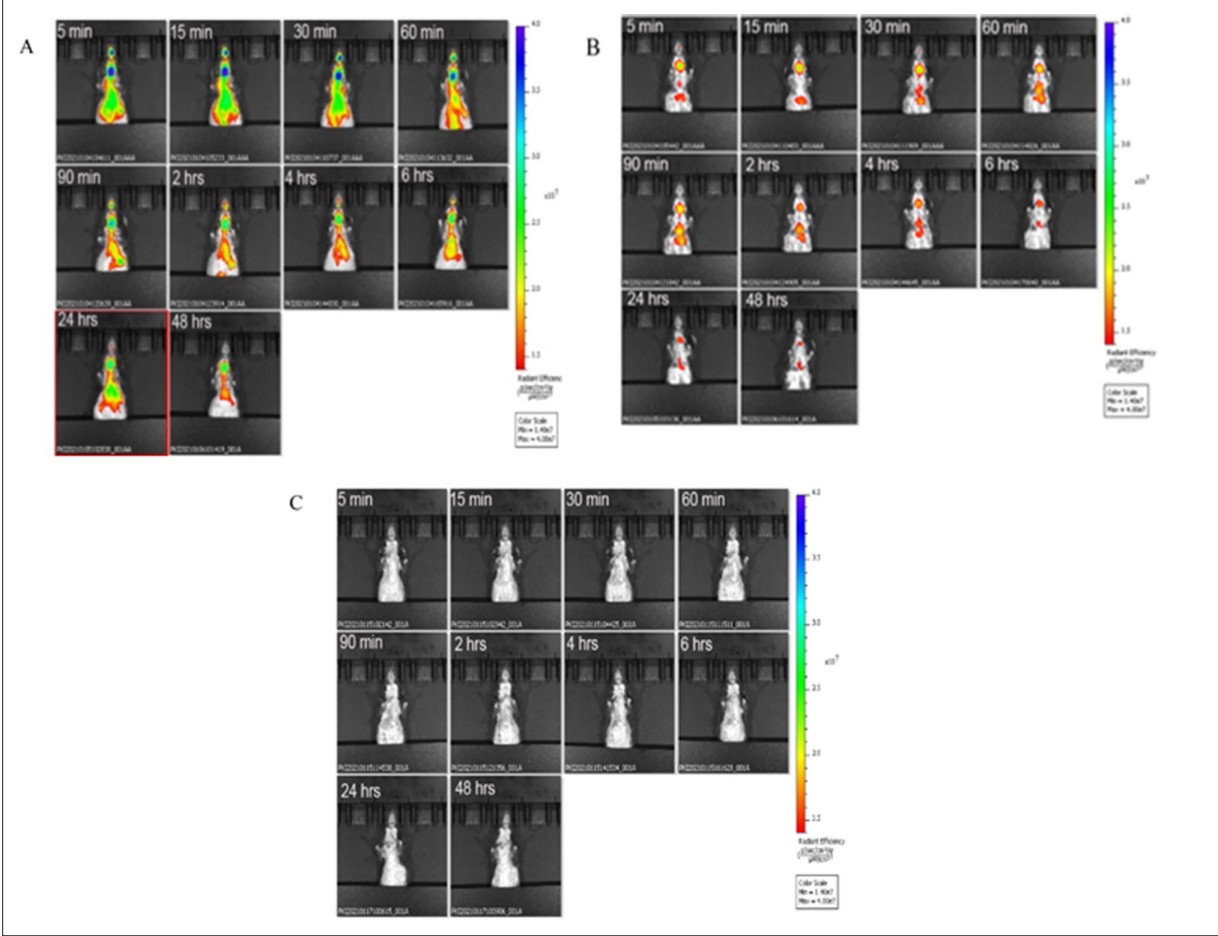

**Figure 4.** Biodistribution of circulating urine-derived exosomes. Shown are representative in vivo bioluminescence images to study the biodistribution of ExoGlow labeled urine-derived exosomes in mice (n=3/group) at the indicated time points. Panel A=mouse injected with labeled U-IPF exo; Panel B=mouse injected with labeled urine-derived exosomes from age and sex-matched control individuals without lung disease. Panel C=mouse injected with PBS. Intensity of luminescence seen in bar from lowest (red) to highest (blue). n=3 individual exosome preparations/group.

The online version of this article includes the following figure supplement(s) for figure 4:

**Figure supplement 1.** Lung fluorescence intensity over time of mice injected with urine-derived exosomes from individuals with IPF (U-IPFexo), urine-derived exosomes from individuals without IPF (Control exosomes), and PBS.

control exosomes (*Figure 5A-C*) or U-IPFexo (*Figure 5D-H*). TEM revealed increased collagen bands in those punches injected with U-IPFexo (*Figure 5H*). Collagen bands were not as evident in the control punches as compared with U-IPF samples due to sampling limitation; however, we observed collagen bundles in both normal and U-IPF-treated samples. Lung structure was preserved in both normal and U-IPF treated samples.

Immunofluorescent staining of punches appeared to have decreased surfactant protein C positive cells (AEC II) after treatment with IPF derived exosomes (*Figure 6C*) compared to treatment with control exosomes (*Figure 6B*). Taken together these data suggest that exosomes are taken up by the alveolar cells and that IPFexo may promote disease by inhibiting the reparative AEC II cells.

## U-IPFexo stimulate a fibrotic pathway response in human lung punches ex vivo

To determine the effects of exosomes on control human lung tissue, we performed ex vivo punch experiments utilizing lungs deemed unsuitable for transplant due to trauma (n=2; *Figure 7*, *Figure 7—source data 1*). $\alpha_v$-integrin and collagen type I mRNA increased after injection with MF-IPFexo compared to control (*Figure 7A*). Consistent with our published data in the lungs of mice with Bleo-injury and lung tissue from individuals with IPF (*Elliot et al., 2019*; *Rubio et al., 2018*), Cav-1 protein

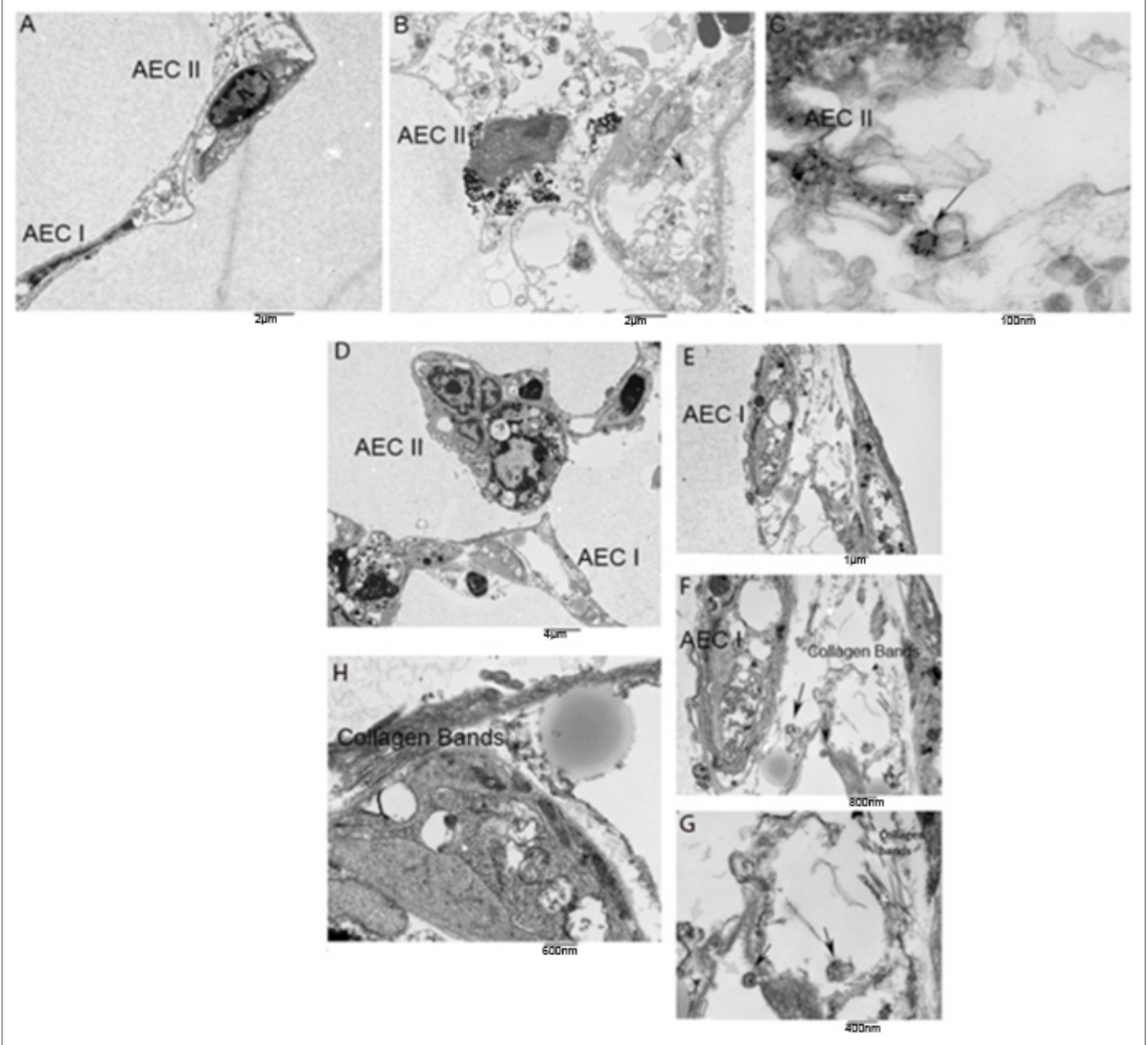

**Figure 5.** Representative TEM photos of lung punches. Panels A-C show mouse lung punches injected with gold nanoparticle labeled urine-derived exosomes from age and sex-matched control subjects (without lung disease) or U-IPFexo (panels D-H). TEM revealed exosomes in alveolar epithelial cells (AEC) type I and type II. Arrows in panels C, F, and G highlight exosomes containing nanoparticles. n=2 individual exosome preparations/group.

The online version of this article includes the following figure supplement(s) for figure 5:

**Figure supplement 1.** Histology and trichrome staining of lung punches from C57BL6 mice.

decreased (*Figure 7B*, *Figure 7—source data 1*) and ERα protein expression increased (*Figure 7C*, *Figure 7—source data 1*) after U-IPFexo were injected into human lung punches. We conducted parallel experiments using lung tissue obtained from 18-month-old male C57BL/6 mice (n=3 technical replicates/individual exosome isolate). Markers for fibrosis ($\alpha_v$-integrin, collagen type I mRNA expression) and other downstream fibrotic pathways (ERα, c-Jun, AKT expression and MMP-9 activity) were only activated by the U-IPFexo and not control exosomes or PBS alone (*Figure 7D–I*, *Figure 7—source data 1*). We found a 2.5 fold increase in TGFβ mRNA expression in punches that were injected

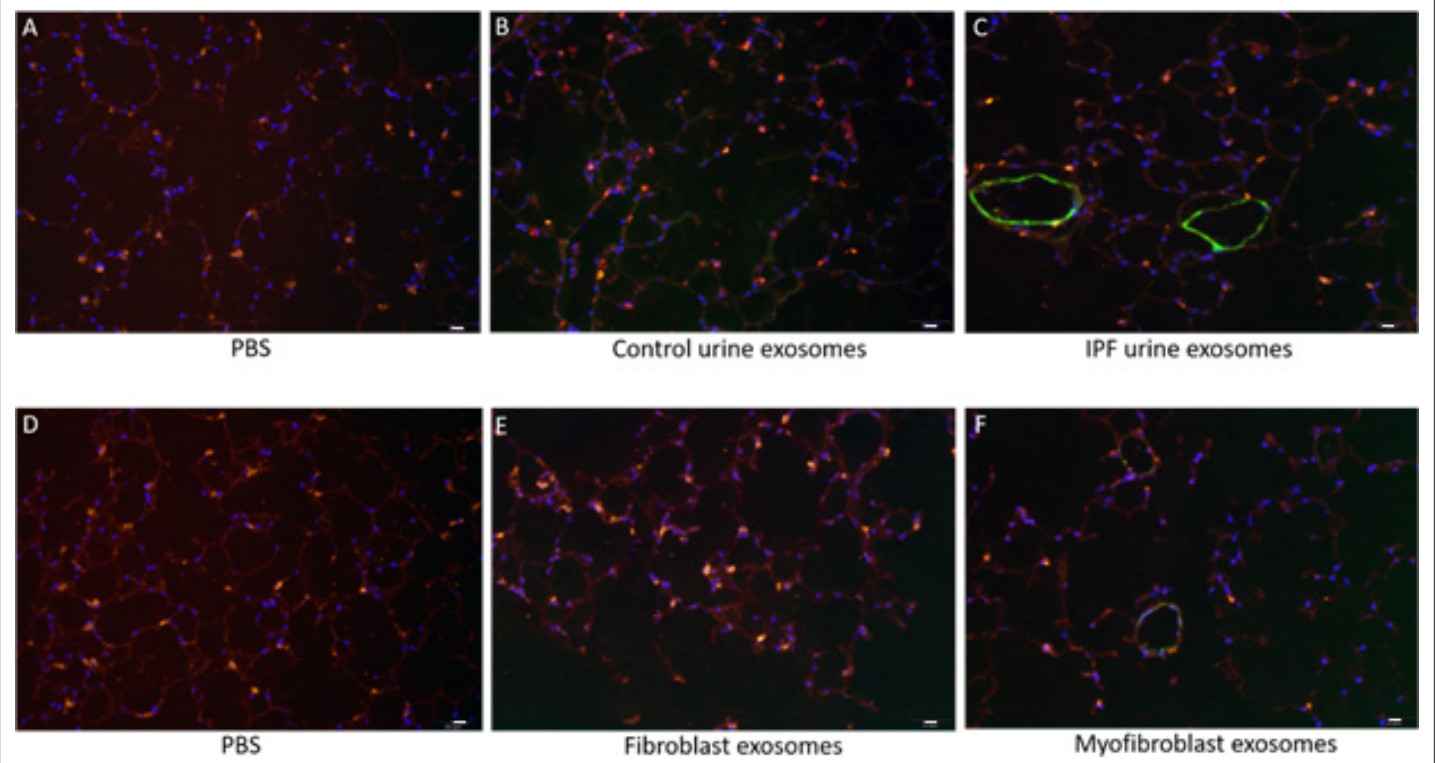

**Figure 6.** Immunofluorescence staining of lung punches injected with exosomes derived from urine (**B–C**) or myofibroblasts or fibroblasts (**E–F**). Lung punches were fixed four days post injection with either PBS (panels **A or D**) or control urine-derived exosomes (panel B), U-IPFexo (panel C), control fibroblast (panel **E**) or MF-IPF exosomes (panel **F**). Shown are representative merged photographs at 20 x, surfactant protein C (SPC, red), αSMC actin (green) and DAPI (blue). n=3 individual exosome preparations/group. Scale bar 50 μm.

with IPF exosomes compared to those injected with PBS or control exosome injections (p<0.05). Similar data were obtained from analysis of punches injected with MF-IPFexo (*Figure 7J–M*, *Figure 7—source data 1*). mRNA expression of α$_v$-integrin and collagen type I were increased in punches injected with MF-IPFexo compared with punches injected with age and sex-matched control fibro-blast exosomes (*Figure 7J, K*, *Figure 7—source data 1*). ERα protein expression shown to be upreg-ulated in the lung tissue and myofibroblasts isolated from individuals with IPF (*Elliot et al., 2019*) was increased after injection of U-IPFexo (1.4 fold increase over control, p=0.05) and MF-IPFexo (>3 fold) into lung punches (*Figure 7F and L*, *Figure 7—source data 1*) compared to control exosomes. Finally, increased c-Jun expression was noted after injection of IPF exosomes but not after injection with control exosomes. U-IPFexo stimulate a fibrotic pathway response like that stimulated by MF-IPFexo.

## U-IPFexo impair tissue repair of human skin ex vivo

We used the human ex vivo skin wound healing assay to test whether U-IPFexo would impair closure of of wounds (*Pastar et al., 2016*; *Stojadinovic and Tomic-Canic, 2013*). U-IPFexo treatment delayed would healing mirroring our previous data showing that mice who develop pulmonary fibrosis after treatment with Bleo also demonstrate a delay in wound healing (*Rubio et al., 2018*). We found that U-IPFexo decreased wound closure compared to control exosomes or PBS (27 ± 6.3% vs 14 ± 0.7%, vs 95%, n=3 technical replicates from two to three biological exosome isolates/group, *Figure 8*, *Figure 8—source data 1*), suggesting that exosomes carry a disease phenotype that can impair tissue repair. As expected, due to age of control exosomes we noted less effective wound healing compared to PBS control.

## U-IPFexo regulate miRNA expression and MMP-9 activity

We measured miRNAs in exosome treated lung punches to determine whether expression changes found in the punches correlated with data published from lung tissue/cells isolated from individuals

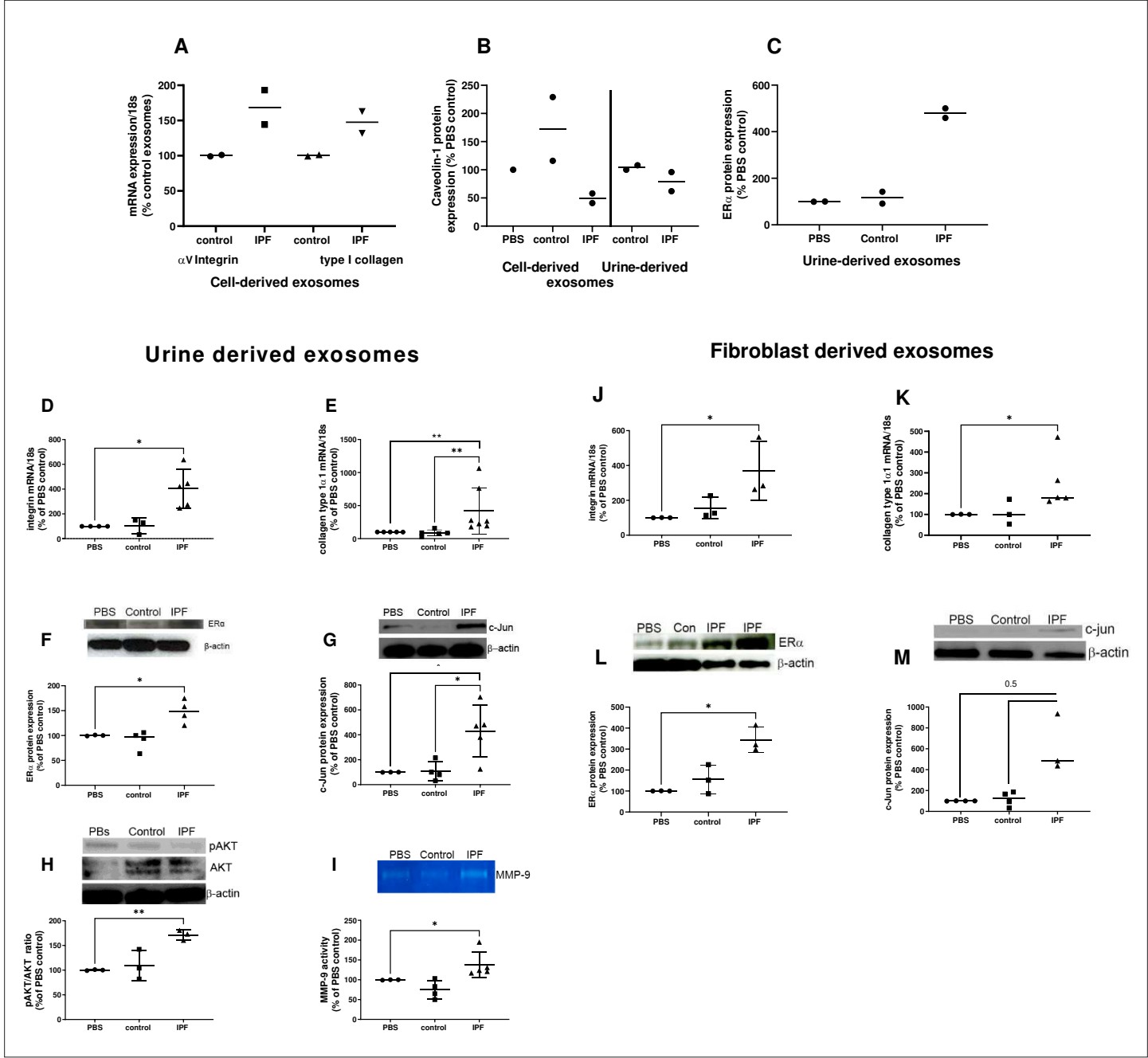

**Figure 7.** Fibrotic pathways are activated in lung punches after injection with urine (U-IPFexo) or myofibroblast-derived (MF-IPFexo) exosomes. Human (**A–C**) and mouse lung (**D–M**) punches were injected with PBS alone, U-IPFexo (**D–I**) or MF-IPFexo (**J–M**) or age and sex-matched control urine exosomes or lung fibroblast exosomes from control subjects (without lung disease). Punches were collected 4 days later and processed as described in Methods. Human lung punches were injected with MF-IPFexo or fibroblast cell derived exosomes (panels A and B) or urine-derived control exosomes and U-IPFexo (panels B and C). n=2 human lung punch isolates, 2 biological exosome preparations/group. Panels D-M, n=3 mouse lung replicates/group, n=3–5 biological exosome isolates/group Data are graphed as percent PBS control. $\alpha_v$-integrin (panels A, D and J, *Figure 7—source data 1*) and collagen type 1 (panels A, E and K, *Figure 7—source data 1*) mRNA expression increased in punches injected with IPFexo (derived from urine or myofibroblasts). Downstream fibrotic pathways; ER$\alpha$ (**C, F and L**), activated AKT (**H**), c-Jun (**G and M**), protein expression and MMP-9 activity (**I**) were also stimulated by exosomes from individuals with IPF. * $p<0.05$, **$p<0.01$. p Values were calculated by Mann Whitney U test.

The online version of this article includes the following source data for figure 7:

**Source data 1.** Raw data for *Figure 7*.

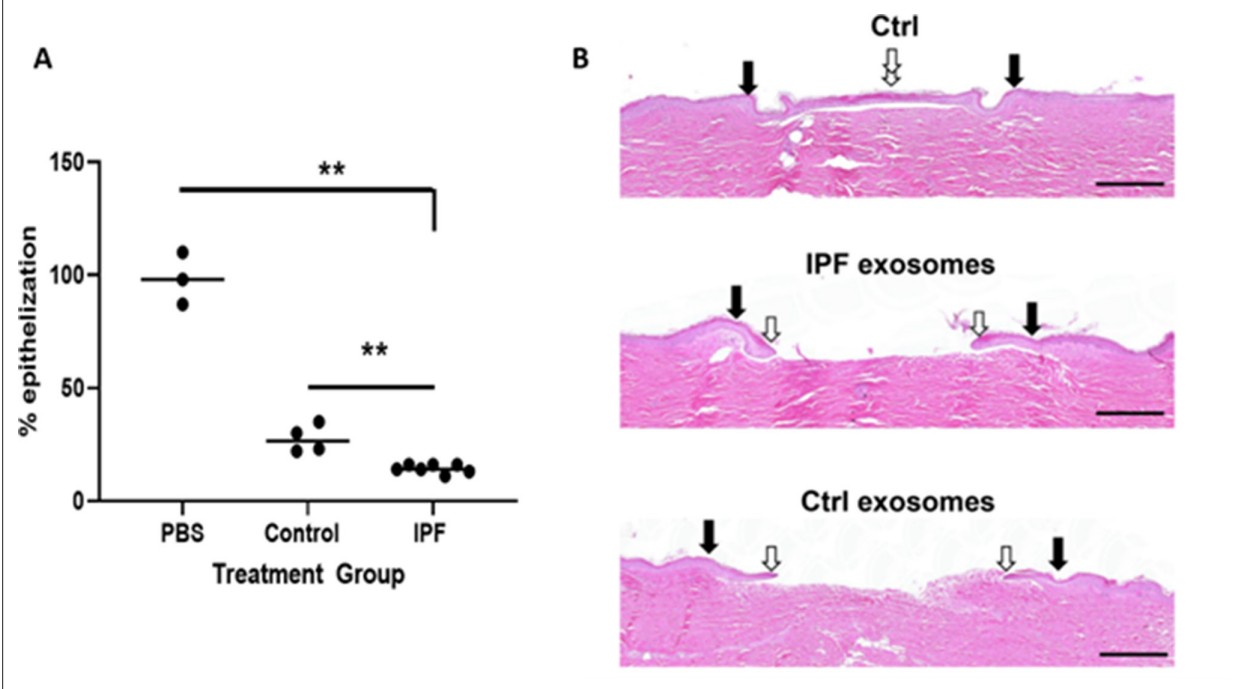

**Figure 8.** Epithelization in ex vivo wound healing is decreased by urine-derived IPF exosomes (U-IPFexo). Human skin was wounded, injected with U-IPFexo or control (age and sex-matched from individuals without lung disease) exosomes and maintained at the air-liquid interface. Wound healing was assessed at day 4 post-wounding, a time point when exponential epithelialization occurs. (**A**) Data are graphed as mean with each data point representing a single wound. Experiments were performed using triplicate technical replicates and two to three biological replicates (*Figure 7—source data 1*). p<0.005 PBS and control compared to IPF, PBS vs control = 0.05 p values were calculated by Mann Whitney U test. (**B**). Photos of gross skin show visual signs of closure and correspond to the histology assessments. Black arrows point to the initial site of wounding, while white arrows point to the wound edge of the migrating epithelial tongue. Scale bars, 500 µm proportional to the image size.

The online version of this article includes the following source data for figure 8:

**Source data 1.** Raw data for *Figure 8*.

with IPF (*Omote and Sauler, 2020*). We found decreased expression of *miR-29* and *miR-let-7d* in punches treated with U-IPFexo (*Table 4*) or MF-IPFexo. MiR-199 was upregulated in those punches injected with U-IPFexo, but remained unchanged by MF-IPFexo. Most likely related to the heterogeneity of the myofibroblasts (*Tsukui et al., 2020*). We and others have previously shown an increase in *miR-199* expression in IPF lung tissue and cells and that increased *miR-199* expression correlates with

**Table 4.** Mouse lung punch miRNA expression.

| Mouse lung punch miRNA expression (% of PBS) | miR-let-7d | miR-29 | miR-181b | miR-199 fibrotic | miR-34a fibrotic | miR-142 antifibrotic |
|---|---|---|---|---|---|---|
| Urine exosomes (n=6–8/group) | | | | | | |
| PBS | 100±0.5 | 100±0.3 | 100±0.2 | 100±0.1 | 100±.3.1 | 100±0.2 |
| Control exosomes | 74±6.7 | 96±13.3 | 137±18 | 92±5 | 88±15 | 105±20 |
| U-IPFexo | 32±6.5 @ | 55±12.1 *& | 52±17 [+] | 169±18 [+] & | 131±10 *[+] | 74±14 [= 0.5] |
| Fibroblast exosomes (n=3–4/group) | | | | | | |
| PBS | 100±0.1 | 100±0.3 | 100±0.5 | 100±0.5 | 100±0.2 | 100±0.2 |
| Control exosomes | 114±5 | 110.8±9.2 | 110±8.2 | 53±5 | 101±16.5 | 78±14 |
| MF-IPFexo | 57±5.9 @ | 56±7.9 *& | 42±12.7@ | 115±14 [NS] | 100.±27 [NS] | 47±12 [NS] |

*+p = 0.05 compared to PBS, @<0.01 compared to PBS and control exosomes, * p<0.05 compared to control exosomes, &p<0.01 compared to control exosomes, p values were calculated by Mann-Whitney *U* test. NS = not significant.

downregulated Cav-1 expression (*Aranda et al., 2015*; *Rubio et al., 2018*). Additional miRNAs were assessed that were reported to be dysregulated in tissues, serum or sputum of individuals with IPF. Punch expression of *miR-34a,* and *miR-142* was regulated in a manner similar to expression found in isolated exosomes after treatment with either U-IPFexo or MF-IPFexo, supporting the relevance of the cargo carried in urine of individuals with IPF (*Shetty et al., 2017*).

## U-IPFexo impact lung fibrosis in vivo

Analysis of lung tissue obtained from Bleo treated mice sacrificed at day 21 showed a higher Ashcroft score (*Figure 9M*, *Figure 9—source data 1*) and collagen content (*Figure 9N*, *Figure 9—source data 1*) in those mice receiving U-IPFexo compared to control exosomes or Bleo alone. $\alpha_V$-integrin (*Figure 9O*, *Figure 9—source data 1*) mRNA expression was higher in lungs of mice that received U-IPFexo compared to Bleo alone. We noted higher *TGFB mRNA* expression in the lung tissue of mice injected with U-IPFexo compared to Bleo treatment alone ($p > 0.001$, $23.8 \pm 3.9$ vs $11.66 \pm 0.6$). Lungs obtained from mice treated with urine exosomes from either asthmatics or subjects with non-CF bronchiectasis also exhibited increased *TGFB mRNA* expression. MMP-2 was induced 2.5-fold by U-IPFexo ($p < 0.05$) compared to Bleo treatment, control exosomes or other exosomes. When Bleo injury was followed by injection of U-IPFexo, fibrosis was more homogeneous and widespread compared to Bleo treatment alone. Mice treated with Bleo +IPF exosomes lost $35 \pm 2.6\%$ BW compared to mice treated with Bleo ($29 \pm 2.6\%$ of BW; $p = 0.05$ compared to IPF) or mice receiving Bleo +control exosomes ($19 \pm 3.2\%$ of BW; $p < 0.05$ compared to Bleo, $p < 0.01$ compared to IPF). Bleo-treated mice that received urine derived exosomes from individuals with non-CF bronchiectasis lost more weight ($39 \pm 5.7\%$ BW; $p < 0.05$) than Bleo-treated mice. The Bleo-treated mice receiving urine derived exosomes from individuals with asthma lost a similar amount of weight as the Bleo-treated mice ($27 \pm 3.5\%$ of BW). We also found that *mir-let-7d,* and *–142* expression decreased while *miR-34a,* and *–199* were increased in lung tissue from Bleo mice treated with U-IPFexo (*Table 5*) but not urine exosomes derived from either asthmatics or subjects with non-CF bronchiectasis (*Table 5*). Although Ashcroft scoring and histology did not demonstrate changes consistent with fibrosis in lung tissue collected from naive animals treated with PBS or naive animals treated with U-IPFexo, increased collagen content (*Figure 9— figure supplement 1*, *Figure 9—figure supplement 1—source data 1*), and dysregulated *miR-34a, miR-199 miR-29*, and *–142* expression (*Table 5*), were detected in lung tissue obtained from these mice.

## Discussion

EVs isolated from various tissues, cells and body fluids contain bioactive molecules such as proteins, lipids, and RNAs derived from the cell of origin (*Collino et al., 2010*; *Vizoso et al., 2017*). Most recently, studies on exosomes have focused on miRNAs, small noncoding RNA molecules able to influence protein expression. Since exosomes act as an interface for cell-cell communication, we investigated IPF disease exosomes and the ability of these 'diseased' exosomes to promote fibrotic lung disease and impair skin wound healing. Our study is the first to report that exosomes isolated from urine of IPF patients, U-IPFexo, confer a disease phenotype in vivo, in human ex vivo lung punches *and* impair ex vivo wound healing in mouse and human tissues. In a similar manner, MF-IPFexo activated markers of fibrosis in ex vivo lung punches compared to punches treated with lung fibroblast-derived exosomes isolated from the lung tissue of age and sex-matched individuals without fibrotic lung disease.

Changes in miRNAs have been implicated in gene expression associated with the development of IPF (*Njock et al., 2019*; *Pandit and Milosevic, 2015*; *Yang et al., 2015*). We found dysregulated expression of miRNAs; *miR-let-7, miR-29a, miR-181b,* and *miR-199* in the isolated U-IPFexo compared to control exosomes. These miRNAs are known to regulate expression of pro-fibrotic, inflammatory and ECM encoding genes (*McDonough et al., 2019b*; *Minnis et al., 2015*; *Njock et al., 2019*; *Noetel et al., 2012*; *Rajasekaran et al., 2015*; *Wang et al., 2016*) and found in lung, serum, and sputum of individuals with IPF (*Elliot et al., 2019*; *McDonough et al., 2019a*; *Pandit et al., 2010*; *Pandit and Milosevic, 2015*; *Rajasekaran et al., 2015*; *Yang et al., 2015*), (*Figure 10*).

*Micro RNA let-7* regulates ERα protein expression as well as the regulation of downstream fibrotic inducers including TGF-β cell signaling pathways (*Elliot et al., 2019*). Dysregulation of *miR-let-7*

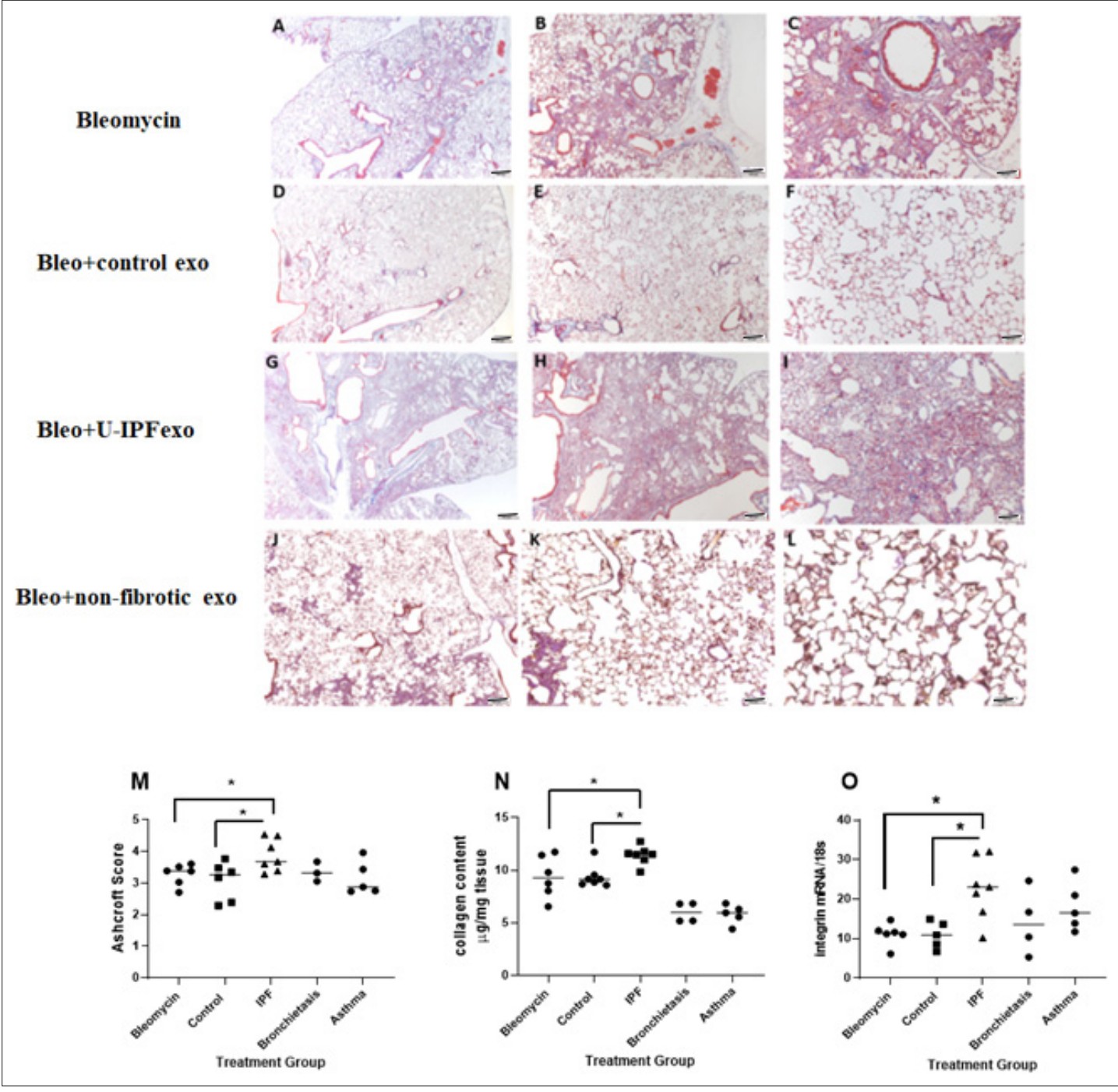

**Figure 9.** Assessment of fibrosis in Bleomycin (Bleo) treated mice intravenously infused with exosomes derived from the urine of individuals with IPF (U-IPFexo) compared to infusion with urine exosomes derived from age and sex- matched control subjects without lung disease or urine exosomes derived from subjects with non-CF bronchiectasis or asthma (non-fibrotic lung disease). Histological sections of lung tissue were stained with Masson's-Trichrome as described in Materials and Methods. Representative photomicrographs (4 x, 10 x, and 20 x) of lung sections from Bleo +vehicle (panels A-C), Bleo +control exosome injected mice (panels D-F), from Bleo +U-IPFexo injected mice (panels G-I) or from non-fibrotic inducing exosomes (Bronchiectasis, panels J-L). Fibrotic score (**M**), collagen content (**N**), $\alpha_v$integrin (**O**) increased after Bleo +U-IPFexo treatment. (**M**) Ashcroft scores were used to evaluate the degree of fibrosis. Data are graphed as the mean score of 32 fields/section of lung. (**N**) Collagen content was estimated by hydroxyproline assay as described in Methods. Data are graphed as µg/mg of lung tissue. (**O**) $\alpha_v$-integrin mRNA expression was determined by RT-PCR as a marker of fibrosis. Data are graphed normalized for 18 S content. Each data point represents an individual mouse, n=4–11 technical replicates/group and two biological replicates/group (*Figure 9—source data 1*) *p<0.05 compared to control exosome treatment or compared to Bleo +vehicle treatment. Data were

*Figure 9 continued on next page*

*Figure 9 continued*

analyzed using one-way analysis of variance (ANOVA) and Mann-Whitney U test. Scale bar panels A, D, G, J, 200 μm; panels B, E,H, K,100 μm; panels C, F,I,L, 50 μm.

The online version of this article includes the following source data and figure supplement(s) for figure 9:

**Source data 1.** Raw data for *Figure 9*.

**Figure supplement 1.** Collagen content increases in mice receiving urine derived exosomes from individuals with IPF.

**Figure supplement 1—source data 1.** Raw data for *Figure 9—figure supplement 1*.

contributes to endothelial/epithelial to mesenchymal transition (Endo/EMT) in the lung (*Pandit and Milosevic, 2015*; *Rajasekaran et al., 2015*), heart (*Wang et al., 2016*) and kidney (*Srivastava et al., 2020*), often leading to fibrosis. Njock et al reported a positive correlation between diffusing capacity of the lungs for carbon monoxide/alveolar volume (DLCO/VA) and presence of let-7d (*Njock et al., 2019*). Dysregulation of miR-29 in the lung alters expression of MMP-2 and collagen1α1, while miR-199 regulates caveolin-1 (*Cushing et al., 2011*; *Montgomery et al., 2014*; *Pandit and Milosevic, 2015*; *Rubio et al., 2018*; *Yang et al., 2013*). Rescue of *miR-181b*, an inhibitor of *NF-κB* targets, suppressed *TGFB* in burns and in a Bleo model of lung injury (*Song and Li, 2015*; *Sun et al., 2012*). Increasing *miR-181b* expression mitigated *TGF-B1* induced EMT in vitro and alleviated alveolar septal thickening and decreased collagen and MMP expression in vivo (*Song and Li, 2015*).

Collectively these data suggest the exosomes have an inherent ability to deliver expression of dysregulated miRNAs and may promote a disease phenotype. When distinct regions of lung tissue were analyzed from the same IPF lung and the histology labeled as minimal, moderate and end-stage disease severity divergent sets of genes and miRNA expression were dependent on the severity of diseased tissue (*McDonough et al., 2019a*). *MicroRNA let-7*, a critical marker in our studies, was associated with end-stage fibrotic lung disease. In a recent study, PBMC miRNA expression correlated with survival in individuals with IPF (*Casanova et al., 2021*), supporting the disease conferring role of miRNA expression as a potential signature of disease. In the present study, we compared expression of miRNAs from urine-derived exosomes and serum-derived exosomes isolated from the same individuals with IPF. We acknowledge that the absence of differences in miRNA expression between urine and serum-derived exosomes may reflect our small sample size. These data warrant a larger study that is currently ongoing.

To demonstrate delivery of exosomes to the lungs, we performed biodistribution experiments using U-IPFexo and urine exosomes from control subjects (without lung disease). The urine-derived exosomes were located in the lungs of 18-month-old male mice within 5 min followed by a diminished signal at 24–48 hr; similar transit time was recorded whether the urine exosomes were prepared from individuals with IPF or controls. Additional studies showed that an exosome signal was no longer visualized in the mouse at 19 days post-injection. Our studies used ExoGlow-Vivo that employs an

**Table 5.** Lung tissue microRNA expression.

| Lung tissue microRNA expression/U6 | miR-let-7d | miR-29 | miR-181b | miR-199 fibrotic | miR-34a fibrotic | miR-142 antifibrotic |
|---|---|---|---|---|---|---|
| PBS | 0.18±0.03 | 2.1±0.4 | 0.26±0.06 | 0.03±0.001 | 0.07±0.02 | 0.23±0.016 |
| Control U-exo | 0.3±0.01 ## | 2.8±0.7 | 0.3±0.07 | 0.06±0.003 | 0.09±0.01 | 0.19±0.03 |
| U-IPFexo | 0.14±0.02 | 0.48±0.1 *& | 0.15±0.05 | 0.05±0.006 † | 0.13±0.03 * | 0.16±0.02 * |
| Bleomycin (Bleo) | 0.057±0.009 ^^ | 0.17±0.02 | 0.009±0.001 | 0.19±0.02 | 0.18±0.05 | 0.23±0.04 |
| Bleo +Control U- exo | 0.24±0.05 † | 0.17±0.02 | 0.008±0.001 | 0.35±0.11 | 0.18±0.02 | 0.29±0.05 |
| Bleo +U-IPFexo | 0.10±0.011 $$^^ | 0.20±0.17 | 0.004±.001 † | 1.68±0.44 *$$^^ | 0.43±0.08 *$$^ | 0.16±0.01 &$^ |
| Bleo +U-bronchiectasis exo | 0.17±.03 | 0.14±0.03 | ND | 0.13±0.35 | 0.06±0.02 | 0.3±0.07 |
| Bleo +U-asthma exo | 0.39±0.1 | 0.11±0.01 | ND | 0.06±0.01 | 0.06±0.02 | 0.29±0.04 |

* $p<0.05$ compared to control exosomes, &$p<0.001$ compared to control exosomes.

† $p<0.05$ compared to Bleo, ##$p<0.001$ compared to Bleo, \$$p<0.05$, \$\$ $p<0.01$ compared to bronchiectasis exosomes ^$p<0.01$ ^^ $p<0.001$ compared to asthma exosomes. p values were calculated by Mann-Whitney *U* test. ND = not detected.

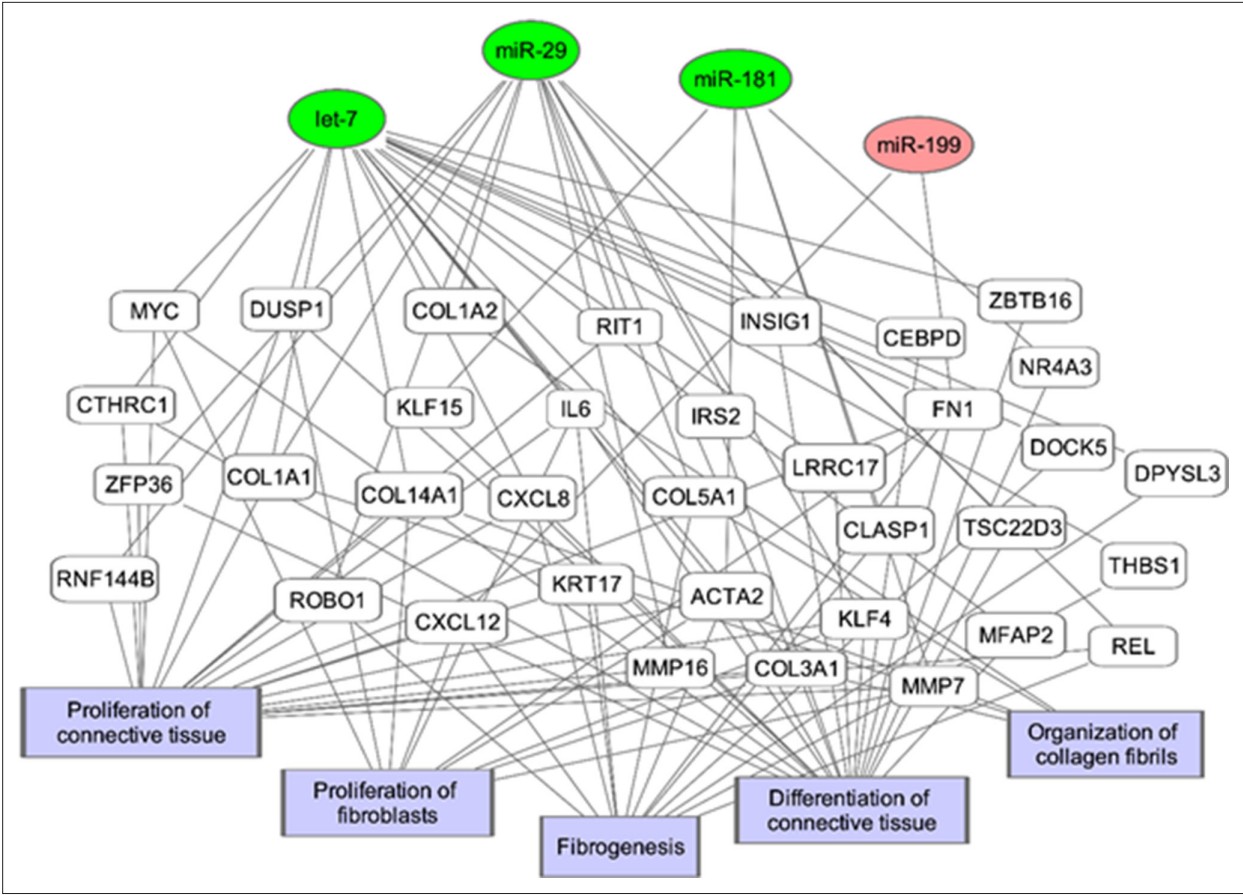

**Figure 10.** Potential microRNA regulated pathways leading to fibrosis. The genes and biological processes in the network are generated from the IPF vs Control Lung dataset from NCBI GEO (GS21369) of 11 IPF samples and 6 healthy lung samples.

amine binding dye emitting in the near infrared (NIR) range instead of infrared lipophilic cyanine dye, DiIC18 (DiD) to label urine derived exosomes. Since the ExoGlow dye is non-lipophilic it provides a more specific signal when used in vivo. Similar transit times have been reported using DiD to label bone marrow mesenchymal stem cell- derived exosomes showing delivery after intravenous infusion to the lung at 5 min and to the liver and spleen at 24 hr (*Grange et al., 2014*). Exosomes derived from breast cancer cells have a similar profile of biodistribution: lung,>liver > spleen, kidney >heart > bone marrow (*Wen et al., 2016*).

To study cellular and tissue interplay in the lung (*Uhl et al., 2015*; *Zscheppang et al., 2018*) we evaluated how the exosomes integrated in ex vivo lung punches by loading exosomes with nanoparticles and performing TEM (*Betzer et al., 2017*; *Zeringer et al., 2013*). We found that the punch architecture was preserved after injection of exosomes compared to non-injected and PBS injected punches, and urine-derived exosomes were integrated into AEC I or AEC II cells. Our studies could not distinguish whether homing of the 'diseased' exosomes favored a specific cell type compared to injection with control exosomes. These extensive studies are ongoing in our laboratory.

Further studies using ex vivo lung punches from human non-fibrotic lung tissue and naïve 18-month-old male mice noted minimal effect on the histology of the lung punch after treatment with PBS or urine-derived exosomes isolated from age and sex-matched controls without lung disease. We found that treatment with U-IPFexo increased expression of fibrotic markers and pathways ($\alpha_v$-integrin, collagen type I and TGFβ mRNA expression, c-Jun protein expression and MMP-9 and AKT activation). Additionally, there was a notable decrease in SPC positive cells in punches (AEC II cells) receiving U-or MF-IPFexo. Since AEC II are regarded as the progenitor population of the alveolus responsible for injury repair and homeostatic maintenance, these data suggest that IPFexo may be inhibiting the repair mechanism in AEC II.

We have previously reported an increase of ERα protein expression in cells and tissue isolated from lungs of male individuals with IPF (*Elliot et al., 2019*). Studies suggest that activation of ERα may be profibrotic in multiple organs including the lung (*Pedram et al., 2010*; *Zhao et al., 2018*). In the current investigation, we found at least ~1.4-fold increase in ERα protein expression in human and mouse punches treated with U-IPFexo or MF-IPFexo. Punches injected with MF-IPFexo also induced other components of fibrotic pathways consistent with data derived from punches injected with U-IPFexo. Furthermore, U-IPFexo impaired the wound healing response, supporting our hypothesis that exosome cargo delivered signals impairing wound repair in skin. This suggests that the systemic feature of IPF exosomes may predispose impaired tissue healing, which can have significant clinical implications and will be further investigated.

We recognize that under our experimental conditions, lung punches lack immune recruitment, systemic perfusion and are under potentially hypoxic conditions. Therefore, we sought to determine if these exosomes would confer lung injury in vivo under normoxic conditions (21% oxygen) in old male C57BL/6 mice. We injected U-IPFexo into naive 18-month-old C57BL/6 mice and collected the lungs 21 days post-exosome treatment. Although there was no change in Ashcroft score, we noted increased collagen content (measured by hydroxyproline) and evidence of inflammatory cells including increased macrophages in lung tissue from mice treated with U-IPFexo compared to tissue from mice treated with control exosomes.

Another group of mice were treated with Bleo. Ashcroft score, collagen content, $\alpha_V$.integrin and TGFβ mRNA expression increased in the lungs of mice treated with Bleo +U-IPFexo compared to mice treated with Bleo +vehicle and Bleo +control exosomes, suggesting that U-IPFexo escalated fibrotic pathways induced by Bleo in old male mice. There was no increase of TGFβ in lungs of mice treated with non-fibrotic exo. As reviewed, TGFβ is ubiquitous in chronic inflammatory lung diseases including COPD, IPF and asthma and may contribute to an immune-suppressed state (*Thomas et al., 2016*).

We also found an expected increase in MMP-2 activity in the lungs of U-IPFexo treated mice, as MMP-2 increases migration and invasiveness of lung myofibroblasts during Bleo-induced pulmonary fibrosis (*Singh et al., 2017*). This was in contrast to urine derived control exosomes or exosomes from urine of individuals with bronchiectasis or asthma. BW, historically shown to decrease with Bleo treatment (*Cowley et al., 2019*) was decreased in mice receiving Bleo +U-IPFexo compared to Bleo alone or Bleo +control exosomes. Loss of BW is associated with worse survival in individuals with IPF (*Mogulkoc et al., 2018*; *Nakatsuka et al., 2018*). Our studies also found similar changes in miRNA expression as reported in prior studies from patients with IPF (*Tzouvelekis et al., 2016*; *Cushing et al., 2011*; *Disayabutr et al., 2016*; *Elliot et al., 2019*; *Guiot et al., 2019*; *Lino Cardenas et al., 2013*; *Njock et al., 2019*; *Omote and Sauler, 2020*; *Ortiz-Quintero et al., 2020*; *Pandit and Milos-evic, 2015*; *Parimon et al., 2019*; *Shetty et al., 2017*; *Yang et al., 2015*).

Our study demonstrates that U-IPFexo preparations contain disease invoking cargo. Parimon et al (*Parimon et al., 2019*) showed that antifibrotic microRNAs (*miRs-144–3 p, −142–3 p, −34b-5p, −503–5* p) packaged into EVs regulate epithelial plasticity and potentially pulmonary fibrosis. Martin-Medina et al (*Martin-Medina et al., 2018*) reported an increase in the number of EV in BALF (bron-choalveolar lavage fluid) from Bleo-treated mice as well as from patients with IPF, that function as carriers for signaling mediators, such as WNT5A. Both studies reported that isolated EV preparations consisted predominantly of exosomes, although microvesicles were present, unlike our preparations that did not contain microvesicles. Neither study investigated urine-derived exosomes. In the present study, lung tissue isolated from naïve or Bleo-treated mice injected with U-IPFexo also exhibited the same changes in miRNA expression as found in ex vivo lung punches. The present study illustrates the potential correlation between changes in mRNA and protein expression (TGFβ, MMPs) and dysregulated miRNA expression suggesting that a miRNA signature could function as a potential biomarker for fibrotic lung disease. Reproducible biomarkers would enable early diagnosis and non-invasive detection methods are currently under investigation for multiple diseases (*Tatler, 2019*).

We further analyzed lungs from Bleo mice that were treated with exosomes derived from subjects with asthma or non-CF bronchiectasis, non-fibrotic lung diseases. We did not find an increase in the fibrotic markers studied and in fact noted a decrease in collagen content and integrin in the lung tissue. A limitation of our current studies includes a limited number of mice which will need to be expanded in future experiments. In addition, we recognize that subjects with asthma were not age matched to the control or IPF subjects. The decreased expression of *miR-34a* and *−199*, fibrotic

inducing microRNAs highlight the differences in cargo content between exosomes from various lung diseases supporting future investigations (*Usman et al., 2021*; *Weidner et al., 2021*).

In an effort to detect expression of other potential fibrotic pathways, we analyzed the urine-exosomes for mRNA expression of *TGFB, IGF, and Cav-1* (*Frangogiannis, 2020*; *Garrett et al., 2019*; *Wicher et al., 2019*). Because exosomes contain non-coding and other small coding RNAs, derived from transfer RNA (tRNA), ribosomal RNA (rRNA), small nuclear RNA (snRNA), and small nucleolar RNA (snoRNA) (*Berrondo et al., 2016*; *Lv et al., 2014*; *Omote and Sauler, 2020*; *Pérez-Boza et al., 2018*), we attempted to characterize the more easily studied expression of mRNA and mRNA fragments (*Batagov and Kurochkin, 2013*; *Peake et al., 2014*). As expected, we found no evidence of mRNA expression for our selected targets (supplement). However, the presence of 3′ UTR mRNA fragments which could bind miRNA and allow for RNA transcription to proceed in target cells ultimately altering the phenotype of otherwise healthy cells has been reported (*Batagov and Kurochkin, 2013*). This is an attractive hypothesis and warrants investigation especially in urine-derived exosomes that elicit a pathologic response (*Batagov and Kurochkin, 2013*). Further studies are in progress to expand our data and include determination of other disease provoking cargo that may be carried by IPF exosomes including tRNAs.

Our study provides the first evidence of circulating cargo found in U-IPFexo that confers disease in two target organs (lung and skin) and suggests a potential mechanism for initiation and/or progression of disease. To the best of our knowledge, the technology to completely empty exosome cargo to allow us to confirm the fibrotic role of specific miRNAs without damaging the exosomes, is not available. Our data suggest that IPF may have a systemic feature. In addition, urine-derived exosomes potentially represent a novel way to identify biomarkers for lung and fibrotic diseases and that miRNA profiling of urine-derived exosomes as biomarkers may lead to noninvasive assessments for earlier diagnosis. Overlap of miRNA expression yielding a fibrotic profile to produce a personalized panel of urine-derived miRNAs or fibromiRs to detect IPF and other lung and fibrotic diseases may be a future diagnostic and prognostic tool.

## Acknowledgements

We acknowledge Vania Almeida and the University of Miami Transmission Electron Microscopy Core for assistance with the generation of EM images. Cheyanne R Head for technical assistance for skin experiments and Joel Fishman MD for help with analysis of patient data.

## Additional information

### Competing interests

Sharon Elliot: holds pending patent applications for Family - Mesenchymal stem cell-derived extra-cellular vesicles and uses thereof for treating and diagnosing fibrotic diseases (30309-001*), Family - Diagnostic and therapeutic uses of compositions comprising purified, enriched potent exosomes containing disease-based and therapy based signature cargo (130309-003*), and Family - Urine-derived exosomes from individuals with IPF carry pro-fibrotic cargo and impair tissue repair (130309-004*). John Ludlow: ZenBio. Sylvia Daunert: participated on a paid role on the Scientific Advisory Board for Akron Biotech and roles on the Scientific Advisory Board for ICN2 and on the Board of Trustees for BIST. SD also received payments/stock options from Berg Pharma and stock options from Aanika Biosciences. The author has no other competing interests to declare. Marjana Tomic-Canic: DSMB, provisional patent, NIH support. Marilyn K Glassberg: holds pending patent applications for Family - Mesenchymal stem cell-derived extracellular vesicles and uses thereof for treating and diagnosing fibrotic diseases (30309-001*), Family - Diagnostic and therapeutic uses of compositions comprising purified, enriched potent exosomes containing disease-based and therapy based signature cargo (130309-003*), and Family - Urine-derived exosomes from individuals with IPF carry pro-fibrotic cargo and impair tissue repair (130309-004*). MKG has a role as Chair, DMSB, Medical College of South Carolina, Mesenchymal Stem Cells in Type I Diabetes (T1D) Phase 1 trial (July 2019-present). The author has no other competing interests to declare. The other authors declare that no competing interests exist.

## Funding

| Funder | Grant reference number | Author |
|---|---|---|
| Lester and Sue Smith Foundation | | Sharon Elliot<br>Marilyn K Glassberg |
| Samrick Family Foundation | | Marilyn K Glassberg |

The funders had no role in study design, data collection and interpretation, or the decision to submit the work for publication.

## Author contributions

Sharon Elliot, Conceptualization, Formal analysis, Supervision, Methodology, Writing – original draft, Project administration, Writing – review and editing; Paola Catanuto, Investigation, Visualization; Simone Pereira-simon, Xiaomei Xia, Investigation; Shahriar Shahzeidi, Evan Roberts, Supervision, Investigation, Methodology, Writing – review and editing; John Ludlow, Validation, Investigation, Methodology, Writing – review and editing; Suzana Hamdan, Investigation, Methodology; Sylvia Daunert, Supervision, Validation, Methodology; Jennifer Parra, Resources, Investigation, Methodology; Rivka Stone, Supervision, Writing – review and editing; Irena Pastar, Conceptualization, Supervision, Funding acquisition, Investigation, Methodology, Writing – review and editing; Marjana Tomic-Canic, Resources, Supervision, Investigation, Methodology, Writing – review and editing; Marilyn K Glassberg, Conceptualization, Resources, Supervision, Funding acquisition, Writing – review and editing

## Author ORCIDs

Sharon Elliot (ID) http://orcid.org/0000-0001-8622-1389

## Ethics

Human subjects: For urine, written informed consent was obtained from the participants with an approved Institutional Review Board protocol (IRB# 20060249) at the University of Miami Miller School of Medicine. Human skin samples were obtained from healthy subjects following panniculectomy (abdominal skin; median age 44 years old). Informed consent was obtained per the requirements of the Institutional Review Board at the University of Miami (IRB protocol # 20070922).

Animal experimentation: All experiments and procedures were approved by the Institutional Animal Care and Use Committee at the Leonard M. Miller School of Medicine at the University of Miami (Miami, FL, protocol number 19-047), a facility accredited by the American Association for the Accreditation of Laboratory Animal Care.

## Decision letter and Author response

Decision letter https://doi.org/10.7554/eLife.79543.sa1
Author response https://doi.org/10.7554/eLife.79543.sa2

# Additional files

## Supplementary files

• MDAR checklist
• Source data 1. Source file for antibodies and primers.
• Source data 2. Raw unedited gels.

## Data availability

Numerical data used to generate the graphic figures is contained in source data files for Figures 2, 6, 7 and 8 and 8*S1.

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
