## [Editor Report]

This study presents valuable findings that exosomes support the development of idiopathic pulmonary fibrosis. The evidence is solid in both in vitro and in vivo bleomycin models of fibrosis. The authors conclude that exosomes might indicate that idiopathic pulmonary fibrosis is a systemic disease.

---

## [Decision Letter]

**Decision letter after peer review:**

Thank you for submitting your article "Urine-derived exosomes from individuals with IPF carry pro-fibrotic cargo" for consideration by eLife. Your article has been reviewed by 3 peer reviewers, one of whom is a member of our Board of Reviewing Editors, and the evaluation has been overseen by Mone Zaidi as the Senior Editor. The following individual involved in the review of your submission have agreed to reveal their identity: Michael Keane (Reviewer #2).

*Reviewer #1 (Recommendations for the authors):*

Overall the paper was nicely written and the experiments were carefully designed to support their aims. But there are a few caveats. First, exosomes carry small cargoes other than RNAs. The authors ruled out the presence of mRNAs in exosomes, but there still could be cargoes other than miRNAs and mRNAs that play pro-fibrotic roles. Second, the authors showed the fibrotic pathways are activated in lung punches. They characterized the expressions of several miRNAs and discussed their roles found in published literature. However, the fibrotic pathways may not be necessarily activated by these miRNAs. MiRNAs are known to have multiple gene targets and are involved in many gene pathways. It would be helpful to include another experiment to show direct connections between these miRNA expression levels and the fibrotic protein levels.

1. It would be helpful to include a demographic feature table and a description of subjects in the method section.

2. Figure 1: Any reason HSP70 is not shown?

3. Figure 2E: Any statistical testing? If from the same individual, paired testing needs to be performed.

4. Figure 5, line 384: Any cell counts per field? How many fewer cells were observed in panel C compared to panel B?

5. Figure 6: Sample size is a little small. Legend is a little confusing. p<0.05 and p<0.01 mentioned twice. Could be more concise. Also, the subtitles size is not consistent. Subtitle for (D-L) should also mention mouse lung punches.

*Reviewer #2 (Recommendations for the authors):*

Elliot et al., describe the presence of profibrotic exosomes in the serum and urine of patients with IPF. Using in vivo models they show that these exosomes traffic to the lung and can be found in alveolar epithelial cells. The authors show that these exosomes support the development of fibrosis both in vitro, using a lung punch model, and in the in vivo bleomycin model. The authors conclude that exosomes may have a role in the development of fibrosis and that they might a indicate that IPF is a systemic disease. The manuscript is interesting and clearly written. There is potential for the development of novel biomarkers from this work.

Why were only exosome from male patients studied?

The authors show that isolated myofibroblasts from IPF lung produce profibrotic exosomes. Do the authors have a sense as to whether exosomes are derived from any other lung cells or indeed any other organs? Is there evidence for similar exosome profile in BALF that might suggest an epithelial source?

The mice used are older than would be conventionally used in the bleomycin model. What is the rationale for using older mice?

Was there any correlation between exosomes and any other clinical parameters such as duration of disease, degree of honeycombing, or 6 minute walk distance?

It would be useful to see the percent predicted values for FEV1/FVC for the subjects enrolled.

The trafficking experiments showing uptake in the lung epithelium are interesting. It would be interesting to know if the pattern of uptake is the same in the bleomycin injured or fibrotic lung.

The non bleomycin (vehicle control) lung is missing from Figure 8. It should at least be included in the hydroxyproline graphs.

The discussion page 39 refers to "the bleomycin model of IPF". While the bleomycin model is an established model of lung injury and fibrosis it is not a model of IPF. This sentence should be reworded.

*Reviewer #3 (Recommendations for the authors):*

The study entitled "Urine-derived exosomes from individuals with IPF carry pro-fibrotic cargo" by Elliot, Glassberg, and colleagues is an interesting and novel proof-of-concept study that aims to determine whether urine derived exosomes from patients with IPF serve as a disease biomarker capable of promoting fibrotic changes. They base their extensive work on the premise that exosomes (derived from multiple sources) in individuals with IPF are key determinants of cell-cell communication that carry aberrant cargo in the form micro-RNAs and stimulate pro-fibrotic pathways upon interaction with receptor cells/tissue. Ultimately, the authors conclude that exosomes isolated from the urine of patients with IPF are indeed capable of conferring a fibrotic phenotype in a manner similar to myofibroblast derived exosomes.

Major Strengths:

– The major strength of this manuscript is that the authors develop/utilize multiple murine and human ex vivo and in vivo models to prove that urine-derive exosomes from patients with IPF confer a pro-fibrotic phenotype. Altogether, they provide a systematic approach in which gold nanoparticles containing exosomes (and their cargo) are administered into the various models and utilize multiple molecular, histologic/microscopy, and biodistribution analyses to substantiate their hypothesis.

– The introduction and discussion are well referenced and provide robust backing for this research.

Major Weaknesses:

– It is uncertain if exosomes derived from control patients are equivalent comparators in their study given that fibroblasts from controls and not myofibroblasts from subjects with IPF are utilized as the source for exosome isolation given that fibroblasts are usually considered quiescent, and they require various stimuli to develop into myofibroblasts. Thus, it is possible that exosomes derived from fibroblasts do not carry similar cargo. In this manner, the comparison that is performed to assess content similarity of exosomes between IPF and controls and between urine and serum is limited to 4 preselected miRNAs, it is likely that high-throughput experiments of exosomal micro-RNA content would aid in determination of similarity/dissimilarity.

– Though the authors developed a systematic workflow that enabled human and murine ex vivo and in vivo validation experiments, there appear to be inconsistencies that raise questions regarding the reproducibility of the experiments that are attributable to different biologic replicate numbers and different stimuli/molecular markers being assessed after different stimuli, in other words experimental design is variable and the modifications to experimental design are not always justified.

– For instance, micro-RNAs 34 and 142 are assessed in the experiments related to Tables 4 and 5 and it is possible that they potentially have biologic relevance; however, these were not reported in the initial exosomal micro-RNA experiment (Figure 2).

– Similarly, there are occasions when myofibroblast and fibroblast derived exosomes may provide valuable information as comparators to substantiate the properties acquired upon administration of urine exosomes.

– Also, exosomes derived from diseased controls (asthma and non-cystic fibrosis bronchiectasis) were obtained but they were utilized as controls only in the in-vivo experiment (Table 5 and Figure 8), an experiment in which markers utilized by the authors as indicative of possible fibrotic stimuli were appreciated in the non-cystic fibrosis bronchiectasis group (TGF-B levels and mouse weight loss).

Justifying the changes in experimental design or providing the additional data to demonstrate consistency would add strength to the findings.

– Many of the experiments are PCR based; however, only relative expression of the molecules of interest is reported and there is no data regarding cycle thresholds at which these were detected. Original RT-qPCR amplification curve and melt curve data would be more informative and it should be provided in the data supplement to substantiate findings and allow for future reproducibility. Along these lines, it is not clear how the findings would change if a more standard analytical approach for normalization of the data were utilized (i.e., ddCT) rather than the normalization against % control exosomes, % PBS control, etc.

– It would serve the manuscript well if the manuscript had standardized terminology/acronyms and figure annotation/labeling throughout.

– The flow of the manuscript methods and Results section is difficult to follow (likely because the work was so extensive), a supplementary figure (flow diagram) would help the reader follow the manuscript's study design.

– Line 123: A reference should be provided to specify the criteria used to determine IPF.

– With regard to the IPF group and the potential to use the urine-derived exosomes as future biomarkers of IPF it would be ideal if Table 1 reports % predicted PFT values (spirometry and DLCO) and it would also be ideal for extent of fibrosis to be reported to gain understanding of disease severity. Since the determination of fibrosis depended on radiologic UIP pattern, there should be more than 1 radiologist interpretation and concordance ascertained (this would allow further substantiation given the small N). What was the time interval between PFT/CT-scan and sample collection for research? Were other causes of UIP pattern in these presumed IPF patients excluded (for instance CT-ILD, often secondary to rheumatoid arthritis , presents with a radiologic UIP pattern)? The importance of this exclusion lies on systemic vs pulmonary specific disease, it is common to think of IPF as a pulmonary only disease whereas RA is a systemic inflammatory disease with joint predominance, as such if the presence of these alternate causes is not excluded it can confound the conclusion that urine exosomes reflect a pulmonary centric disease.

– What were the cell markers to ascertain fibroblasts? The cited study (Elliot 2019) only provides information on myofibroblasts. As stated in the major comments, the question of fibroblast vs myofibroblast validity as a comparison arises from the consideration that it would be unexpected for cells with different transcriptional programs to confer the same cargo to exosomes- with this being said, a high-throughput experiment assessing differences in exosomal cargo between the cell types may prove/disprove this given that the preselected micro-RNAs may not be fully representative of the full exosomal cargo.

– The methods section would likely benefit from section reordering or providing reference to future sections to allow there to be a better understanding of the workflow. For instance, line 162 as written does not make sense given that it states: "exosomes were injected into the lung punch …in a volume of 100uL." It is not until line 200 that it becomes apparent that these are mixed in with the nanoparticles. Other instances require clarification throughout the methods (where do the human lung samples come from? – not mentioned until the Results section, etc.)

– The human lung samples are mentioned to be obtained from trauma patient explants, were these patients previously healthy or were the lungs "primed" for fibrotic processes from underlying disease/or were they diseased and only histologically normal sites utilized?

– Line 199- "Experiments" is misspelled.

– Line 279: clarify, does this line refer to "bleomycin" naïve control mice?

– Histology: Ashcroft scoring would benefit from interpretation from various pathologists and concordance assessment given the small N.

– Table 1 is missing demographic characteristics that may play a role (such as race) and clinical characteristics such as treatment or lack thereof of with antifibrotics. With regards to the asthma and non-cf bronchiectasis groups- were these obtained when patients were flaring or not flaring? In the case of the non-cf bronchiectasis group if a flare was present this could in part explain the effects observed in the mice (weight decreased even more than in the IPFexo administration group).

– Line 356: the statement is too broad knowing that more than the preselected micro-RNAs are carried in these exosomes. Equivalence can only refer to the micro-RNAs that were tested in Figure 2.

– Figure 2: where is the data pertaining to expression of micro-RNA 34 and 142 that become featured micro-RNAs in the latter experiments? What led to the variability in sample numbers/replicates? Figure 2E, what individuals were selected to show equivalent miRNA expression (with the variable replicates/samples in the prior panels) it makes it difficult to draw conclusions- were these selected as a convenience sample, randomly, etc. Moreover, it seems like some subjects are missing from the comparison (some points on the scatter are lacking- this may be from overlap and the "jitter" in Graphpad can be adjusted to show all data points). Also, the statistic here may not be the most appropriate this seems like it calls for a paired-sample assessment rather than and MWU test (i.e Wilcoxon signed rank test for non-normal data). This applies to other instances of paired assessments as well.

– Line 375- missing the word "in": exosomes were located IN the lung.

– Supplemental Figure 1: because 3 mice were used in each condition it would be best to show all 3 data points for each condition at the various time points. A mean value between three data points is not sufficient to show differences given that the variability can be high (ie. 49,50,51) have a mean value of 50, while 0,50,100 would also have a mean value of 50. Panel B caption text refers to both 30 min and 48 hour timepoints but the figure only shows data for 30min timepoint. What does the label "IPF 178" refer to in Panel B?

– Line 405: please clarify, not imaged or not visualized in TEM. Seems that this is showing the absence of collagen deposition is just as relevant to controls as is showing the presence of collagen in U-IPFexo given that a potential causality (or proof of) is sought by these experiments.

– Line 426-427: is an interesting hypothesis linking exosomal cargo to deficiencies in surfactant and AECs that are thought to contribute to fibrotic mechanisms. However, no comment is made on the effect of myofibroblast/fibroblast derived exosomes that are shown in Figure 5E. Are there conclusions to be drawn from these stains as well?

– With regards to U-IPFexo on human lung punches -> there is mention of collagen and integrin being assessed- however, it seems that this is not commented on, what effect did U-IPF exosomes have on collagen and integrin.

– Figure 6- also needs better labeling, for concordance- where is the ERa response to myofibroblast/fibroblast exosomes? Again, there seems to be variability in replicates (how many human lung replicates?, mouse replicates are provided but variable…why the variability?), also if only 2 replicates were assessed individual points and not bar plots with error bars should be shown. Figure 6 caption/labeling do not seem to be consistent- needs revision, axis titles need revision. For visualization purposes- data/replicate summary in caption makes interpretation difficult, some of the caption can be moved to the figure panels themselves, to show pertinent data for each experiment. MWU tests need to be clarified as pairwise comparisons given that the significance bars that are shown are confusing in the current format- significance bars would need revision.

– Line 403: Misspelled "wound" (sentence says delayed would healing).

– Line 487- avoid definitive statements given small N ◊ "confirming that exosomes".

– Line 488 sentence needs to be clarified, is it: control in comparison to PBS control?

– Figure 7: caption states that data is shown as mean+/- SEM though error bars are not shown. Panel B, scale at the bottom right corner is difficult to visualize. Maybe extend from 100um to 1000um with tick marks every 100um.

– With regards to figure 7 (and falling back on the fact that exosomes may contain similar or dissimilar content between IPF and controls)- can a comment be made on why control exosome skin does not heal? Though the difference in % healing is significantly greater than IPF exosomes, there is still a substantial impairment in wound healing.

– Line 502: measured is misspelled.

– U-IPFexo micro-RNA expression/MMP9 activity paragraph- Was the method for micro-RNA extraction from tissue described in the methods/supplement?

– Line 507: Why would cell-specific exosomal micro-RNA content be less heterogenous than circulating/urine (excreted) exosomal micro-RNA content? Given that urine exosomes are unlikely to come from a single source would there not be more heterogeneity in urine?

– Tables 4 and 5 are difficult to interpret because significance symbols and text blend together.

– Is the corresponding experiment for Table 4 in mouse lung, human lung, both? Clarify.

– Line 511-514: Conclusion regarding miR-34 does not seem to be consistent with what is shown in Table 4. Table 4 demonstrates a relative upregulation after urine exo versus a relative and seemingly significant downregulation with myofibroblast exo (though this is labeled as "NS"). Similarly, conclusions relating to miR-142 seem to be discrepant and seemingly significant based on the table data. Would recommend that MWU -tests for this data be revised. Can these discordant relationships be commented on?

– Line 532: Needs a period between the closing parenthesis and Bleo to start the sentence.

– Bleomycin + Control exosomes seem to have a protective effect against fibrosis, is this because of the different cell lines (myofibroblasts vs fibroblasts- with fibroblasts providing regenerative properties to the tissue?)

– Line 529- in terms of body weight what can be said about non-cf bronchiectasis mice losing at least as much body weight as IPFexo mice (again in reference to are the non-cf bronchiectasis mice exosomes carrying "inflammatory" cargo).

– Can a comment be made on the discrepancy between miR-142 in the punch biopsy experiment and the in vivo experiments? In the punch experiment miR-142 remains at stable levels whereas in the in vivo experiment it is decreased.

– Table 5. If ANOVA was performed a post hoc analyses (Tukey) would likely be more suitable.

– For the in vivo experiment: are these data concordant with myofibroblast/fibroblast derived exosome administration? It seems that this would be a relevant experiment to show that there is concordance between urine and cell-specific profibrotic changes.

– Discussion will likely need revision with changes. In line 704- again there is a definitive statement which should be avoided given small N.

Drs. Glassberg and Elliot provided a novel exosome source-urine derived exosome, from IPF and non-IPF patients, as well as controls, as one of the intervention agents to verify their previous reported pro-fibrotic miRNAs by using the skin and lung tissue models in this manuscript. It's also the first time to report the comparison of the contents of exosomes from different sources.

In this manuscript, they reported that the similar lung fibrotic specific markers(miRNAs) expression pattern was observed from both the serum and urine, suggested that the dysregulated miRNAs can be detected systematically, outside of lung. Therefore, the urine derived exosome can potentially be used as biomarkers to define the disease of IPF, meaningful for the noninvasive diagnosis at the early stage, as well as for the prognosis with the molecular level phenotyping the disease of IPF.

Based on the data driven hypothesis, in brief, the "disease" exosomes transport dysregulated miRNAs, into circulating system, which could be responsible for the fibrotic lung tissue, and involved in the development and progression of fibrosis, they designed a systematic workflow to obtain, validate, distribute the dysregulated miRNAs from the fibrotic and control subjects, and then verify their hypothesis on both lung and skin models. The application of in vivo bio-distribution, along with the other robust molecular analyses supports the conclusion solidly. No doubt, it deserves to be published.

---

## [Author Response]

Reviewer #1 (Recommendations for the authors):Overall the paper was nicely written and the experiments were carefully designed to support their aims. But there are a few caveats. First, exosomes carry small cargoes other than RNAs. The authors ruled out the presence of mRNAs in exosomes, but there still could be cargoes other than miRNAs and mRNAs that play pro-fibrotic roles. Second, the authors showed the fibrotic pathways are activated in lung punches. They characterized the expressions of several miRNAs and discussed their roles found in published literature. However, the fibrotic pathways may not be necessarily activated by these miRNAs. MiRNAs are known to have multiple gene targets and are involved in many gene pathways. It would be helpful to include another experiment to show direct connections between these miRNA expression levels and the fibrotic protein levels.

The authors are performing a larger study to assess the direct connections between the miRNAs studied and fibrotic pathways including proteomic analyses. However, we have included a pathway analysis based on the potential interactions of the studied miRNAs.

1. It would be helpful to include a demographic feature table and a description of subjects in the method section.

We have added the information to the tables and to the methods section.

2. Figure 1: Any reason HSP70 is not shown?

We have routinely tested for multiple recognized markers of exosomes including HSP70. We selected only a few for this manuscript.

3. Figure 2E: Any statistical testing? If from the same individual, paired testing needs to be performed.

Paired testing was performed and added to manuscript.

4. Figure 5, line 384: Any cell counts per field? How many fewer cells were observed in panel C compared to panel B?

We appreciate the reviewer’s comment and did not count the cell number. We will accomplish this in our future study.

5. Figure 6: Sample size is a little small. Legend is a little confusing. p<0.05 and p<0.01 mentioned twice. Could be more concise. Also, the subtitles size is not consistent. Subtitle for (D-L) should also mention mouse lung punches.

The legend has been edited as per suggestions.

Reviewer #2 (Recommendations for the authors):Elliot et al., describe the presence of profibrotic exosomes in the serum and urine of patients with IPF. Using in vivo models they show that these exosomes traffic to the lung and can be found in alveolar epithelial cells. The authors show that these exosomes support the development of fibrosis both in vitro, using a lung punch model, and in the in vivo bleomycin model. The authors conclude that exosomes may have a role in the development of fibrosis and that they might a indicate that IPF is a systemic disease. The manuscript is interesting and clearly written. There is potential for the development of novel biomarkers from this work.Why were only exosome from male patients studied?

IPF is a disease predominately found in males over the age of 55. This manuscript focuses on that population. Sex and age exosome cargo derived from urine of these and other subjects is currently under investigation in our laboratory.

The authors show that isolated myofibroblasts from IPF lung produce profibrotic exosomes. Do the authors have a sense as to whether exosomes are derived from any other lung cells or indeed any other organs? Is there evidence for similar exosome profile in BALF that might suggest an epithelial source?

We agree that there are multiple sources of exosomes in the lung, however in this study we focused on the comparison between the urine and the myofibroblast.

The mice used are older than would be conventionally used in the bleomycin model. What is the rationale for using older mice?

Our studies and those of others have found that aging mice that do not repair spontaneously after bleomycin instillation are a more appropriate pre-clinical model. (Tashiro et, Redente et al. and Sueblinvong V).

Was there any correlation between exosomes and any other clinical parameters such as duration of disease, degree of honeycombing, or 6 minute walk distance?

This pilot study was not designed with those parameters. Our larger ongoing study will address these important correlations.

It would be useful to see the percent predicted values for FEV1/FVC for the subjects enrolled.

These values have been added to updated Table 1.

The trafficking experiments showing uptake in the lung epithelium are interesting. It would be interesting to know if the pattern of uptake is the same in the bleomycin injured or fibrotic lung.

Those experiments are ongoing.

The non bleomycin (vehicle control) lung is missing from Figure 8. It should at least be included in the hydroxyproline graphs.

PBS is the vehicle control shown in Figure 8 supplement 1.

The discussion page 39 refers to "the bleomycin model of IPF". While the bleomycin model is an established model of lung injury and fibrosis it is not a model of IPF. This sentence should be reworded.

We have reworded to say bleomycin model of lung injury.

Reviewer #3 (Recommendations for the authors):The study entitled "Urine-derived exosomes from individuals with IPF carry pro-fibrotic cargo" by Elliot, Glassberg, and colleagues is an interesting and novel proof-of-concept study that aims to determine whether urine derived exosomes from patients with IPF serve as a disease biomarker capable of promoting fibrotic changes. They base their extensive work on the premise that exosomes (derived from multiple sources) in individuals with IPF are key determinants of cell-cell communication that carry aberrant cargo in the form micro-RNAs and stimulate pro-fibrotic pathways upon interaction with receptor cells/tissue. Ultimately, the authors conclude that exosomes isolated from the urine of patients with IPF are indeed capable of conferring a fibrotic phenotype in a manner similar to myofibroblast derived exosomes.Major Strengths:– The major strength of this manuscript is that the authors develop/utilize multiple murine and human ex vivo and in vivo models to prove that urine-derive exosomes from patients with IPF confer a pro-fibrotic phenotype. Altogether, they provide a systematic approach in which gold nanoparticles containing exosomes (and their cargo) are administered into the various models and utilize multiple molecular, histologic/microscopy, and biodistribution analyses to substantiate their hypothesis.– The introduction and discussion are well referenced and provide robust backing for this research.Major Weaknesses:– It is uncertain if exosomes derived from control patients are equivalent comparators in their study given that fibroblasts from controls and not myofibroblasts from subjects with IPF are utilized as the source for exosome isolation given that fibroblasts are usually considered quiescent, and they require various stimuli to develop into myofibroblasts. Thus, it is possible that exosomes derived from fibroblasts do not carry similar cargo. In this manner, the comparison that is performed to assess content similarity of exosomes between IPF and controls and between urine and serum is limited to 4 preselected miRNAs, it is likely that high-throughput experiments of exosomal micro-RNA content would aid in determination of similarity/dissimilarity.

We recognize the limitations of our study and ongoing experiments in our laboratory include a substantially increased cohort and sequencing of these samples. The intention of this study was to assess whether those miRNAs known to be dysregulated in the lung, sputum and serum of subjects with IPF would exhibit those differences in urine derived exosomes.

– Though the authors developed a systematic workflow that enabled human and murine ex vivo and in vivo validation experiments, there appear to be inconsistencies that raise questions regarding the reproducibility of the experiments that are attributable to different biologic replicate numbers and different stimuli/molecular markers being assessed after different stimuli, in other words experimental design is variable and the modifications to experimental design are not always justified.

Many of the experiments were limited by the number of isolated exosomes obtained vs the amount needed for ex vivo and in vivo studies which we deemed important for functional analysis.

– For instance, micro-RNAs 34 and 142 are assessed in the experiments related to Tables 4 and 5 and it is possible that they potentially have biologic relevance; however, these were not reported in the initial exosomal micro-RNA experiment (Figure 2).

We consistently find a high CT number for these microRNAs in isolated exosomes. The authors agree this data is important to obtain and will be included in the upcoming larger study.

– Similarly, there are occasions when myofibroblast and fibroblast derived exosomes may provide valuable information as comparators to substantiate the properties acquired upon administration of urine exosomes.

This manuscript only focused on the similarity in cargo of urine, serum and myofibroblast exosomes.

– Also, exosomes derived from diseased controls (asthma and non-cystic fibrosis bronchiectasis) were obtained but they were utilized as controls only in the in-vivo experiment (Table 5 and Figure 8), an experiment in which markers utilized by the authors as indicative of possible fibrotic stimuli were appreciated in the non-cystic fibrosis bronchiectasis group (TGF-B levels and mouse weight loss).Justifying the changes in experimental design or providing the additional data to demonstrate consistency would add strength to the findings.

The authors appreciate the comments of the reviewers and realize that in this pilot study we were unable to measure all parameters in all experiments. We were limited in the amount of urine, and therefore exosome material, obtained from the non-fibrotic exosomes were not analyzed for specific cargo. The manuscript contains the BW change for asthma and bronchiectasis treated mice (see line 530). The change in BW after administration of non-CF bronchiectasis urine derived exosomes may reflect the potential ongoing inflammatory state in these patients. We believe the cargo is different between lung pathologies and is part of the ongoing investigation in our laboratory.

– Many of the experiments are PCR based; however, only relative expression of the molecules of interest is reported and there is no data regarding cycle thresholds at which these were detected. Original RT-qPCR amplification curve and melt curve data would be more informative and it should be provided in the data supplement to substantiate findings and allow for future reproducibility. Along these lines, it is not clear how the findings would change if a more standard analytical approach for normalization of the data were utilized (i.e., ddCT) rather than the normalization against % control exosomes, % PBS control, etc.

Data was derived using the ddCT method as outlined by Livak KJ et al. This data was then transformed into % vehicle or PBS for ease of graphing.

– It would serve the manuscript well if the manuscript had standardized terminology/acronyms and figure annotation/labeling throughout.– The flow of the manuscript methods and Results section is difficult to follow (likely because the work was so extensive), a supplementary figure (flow diagram) would help the reader follow the manuscript's study design.

The authors appreciate the suggestions, and we hope that the manuscript has been improved. We have included a flow diagram for ease of understanding the study design.

– Line 123: A reference should be provided to specify the criteria used to determine IPF.

A reference has been added.

– With regard to the IPF group and the potential to use the urine-derived exosomes as future biomarkers of IPF it would be ideal if Table 1 reports % predicted PFT values (spirometry and DLCO) and it would also be ideal for extent of fibrosis to be reported to gain understanding of disease severity. Since the determination of fibrosis depended on radiologic UIP pattern, there should be more than 1 radiologist interpretation and concordance ascertained (this would allow further substantiation given the small N). What was the time interval between PFT/CT-scan and sample collection for research? Were other causes of UIP pattern in these presumed IPF patients excluded (for instance CT-ILD, often secondary to rheumatoid arthritis , presents with a radiologic UIP pattern)? The importance of this exclusion lies on systemic vs pulmonary specific disease, it is common to think of IPF as a pulmonary only disease whereas RA is a systemic inflammatory disease with joint predominance, as such if the presence of these alternate causes is not excluded it can confound the conclusion that urine exosomes reflect a pulmonary centric disease.

Scans were typically read close to the time of urine collection prior to clinic appointment.

– What were the cell markers to ascertain fibroblasts? The cited study (Elliot 2019) only provides information on myofibroblasts. As stated in the major comments, the question of fibroblast vs myofibroblast validity as a comparison arises from the consideration that it would be unexpected for cells with different transcriptional programs to confer the same cargo to exosomes- with this being said, a high-throughput experiment assessing differences in exosomal cargo between the cell types may prove/disprove this given that the preselected micro-RNAs may not be fully representative of the full exosomal cargo.

We performed immunohistochemistry on the cells using αSMC actin and vimentin antibodies. Since we only expect to see expression of αSMC in “activated” fibroblasts, we used that as our criteria. The authors agree that for this initial pilot investigation we preselected a group of miRNAs that were known to be dysregulated in the lungs, serum of patients with IPF.

– The methods section would likely benefit from section reordering or providing reference to future sections to allow there to be a better understanding of the workflow. For instance, line 162 as written does not make sense given that it states: "exosomes were injected into the lung punch …in a volume of 100uL." It is not until line 200 that it becomes apparent that these are mixed in with the nanoparticles. Other instances require clarification throughout the methods (where do the human lung samples come from? – not mentioned until the Results section, etc.)

We thank the reviewer for the helpful comments and have updated the methods section.

– The human lung samples are mentioned to be obtained from trauma patient explants, were these patients previously healthy or were the lungs "primed" for fibrotic processes from underlying disease/or were they diseased and only histologically normal sites utilized?

None of the lung samples were obtained from patients with underlying comorbidities in their medical records or in imaging.

– Line 199- "Experiments" is misspelled.– Line 279: clarify, does this line refer to "bleomycin" naïve control mice?

Yes, that clarification has been added to the manuscript.

– Histology: Ashcroft scoring would benefit from interpretation from various pathologists and concordance assessment given the small N.

We agree that future studies should utilize multiple pathologists requiring samples being sent to other mouse pathologists. Samples were read by a mouse pathologist.

– Table 1 is missing demographic characteristics that may play a role (such as race) and clinical characteristics such as treatment or lack thereof of with antifibrotics. With regards to the asthma and non-cf bronchiectasis groups- were these obtained when patients were flaring or not flaring? In the case of the non-cf bronchiectasis group if a flare was present this could in part explain the effects observed in the mice (weight decreased even more than in the IPFexo administration group).

Table 1 has been updated. The samples were obtained from patients with stable non-flaring disease.

– Line 356: the statement is too broad knowing that more than the preselected micro-RNAs are carried in these exosomes. Equivalence can only refer to the micro-RNAs that were tested in Figure 2.

We have modified the paragraph to better reflect the select miRNAs that we have chosen to investigate.

– Figure 2: where is the data pertaining to expression of micro-RNA 34 and 142 that become featured micro-RNAs in the latter experiments? What led to the variability in sample numbers/replicates? Figure 2E, what individuals were selected to show equivalent miRNA expression (with the variable replicates/samples in the prior panels) it makes it difficult to draw conclusions- were these selected as a convenience sample, randomly, etc. Moreover, it seems like some subjects are missing from the comparison (some points on the scatter are lacking- this may be from overlap and the "jitter" in Graphpad can be adjusted to show all data points). Also, the statistic here may not be the most appropriate this seems like it calls for a paired-sample assessment rather than and MWU test (i.e Wilcoxon signed rank test for non-normal data). This applies to other instances of paired assessments as well.

The comparison of urine vs serum was performed on the samples that we obtained initially from clinic patients. Further collection of urines did not always include serum collection. Figure 2 has been revised to show only single samples that were processed. In some cases, urine samples were aliquoted, frozen and variability assessed in several of the frozen aliquots. Those duplicates were removed from the graphs. The authors agree that paired sample testing should be used for urine vs serum data. This has been included in the revised version of the manuscript.

– Line 375- missing the word "in": exosomes were located IN the lung.

Corrected

– Supplemental Figure 1: because 3 mice were used in each condition it would be best to show all 3 data points for each condition at the various time points. A mean value between three data points is not sufficient to show differences given that the variability can be high (ie. 49,50,51) have a mean value of 50, while 0,50,100 would also have a mean value of 50. Panel B caption text refers to both 30 min and 48 hour timepoints but the figure only shows data for 30min timepoint. What does the label "IPF 178" refer to in Panel B?

The authors have corrected the results and figure and added three time points on the graphs

– Line 405: please clarify, not imaged or not visualized in TEM. Seems that this is showing the absence of collagen deposition is just as relevant to controls as is showing the presence of collagen in U-IPFexo given that a potential causality (or proof of) is sought by these experiments.

We recognize that punch biopsy cannot be used for quantification of collagen deposition in the matrix therefore we have modified the sentence in the manuscript.

– Line 426-427: is an interesting hypothesis linking exosomal cargo to deficiencies in surfactant and AECs that are thought to contribute to fibrotic mechanisms. However, no comment is made on the effect of myofibroblast/fibroblast derived exosomes that are shown in Figure 5E. Are there conclusions to be drawn from these stains as well?

The figure demonstrates comparable results in both populations of exosomes supporting their origin from the same disease background.

– With regards to U-IPFexo on human lung punches -> there is mention of collagen and integrin being assessed- however, it seems that this is not commented on, what effect did U-IPF exosomes have on collagen and integrin.

We were limited in our ability to show the effect on human lung due to the amount of lung tissue available. Therefore, we completed those experiments using mouse lung.

– Figure 6- also needs better labeling, for concordance- where is the ERa response to myofibroblast/fibroblast exosomes? Again, there seems to be variability in replicates (how many human lung replicates?, mouse replicates are provided but variable…why the variability?), also if only 2 replicates were assessed individual points and not bar plots with error bars should be shown. Figure 6 caption/labeling do not seem to be consistent- needs revision, axis titles need revision. For visualization purposes- data/replicate summary in caption makes interpretation difficult, some of the caption can be moved to the figure panels themselves, to show pertinent data for each experiment. MWU tests need to be clarified as pairwise comparisons given that the significance bars that are shown are confusing in the current format- significance bars would need revision.

The figures and legend have been edited per the reviewer’s comments. Accompanying Supplementary files and a new graph were added for urine exosome and ERα expression response.

– Line 403: Misspelled "wound" (sentence says delayed would healing).– Line 487- avoid definitive statements given small N ◊ "confirming that exosomes".– Line 488 sentence needs to be clarified, is it: control in comparison to PBS control?

The above corrections have been made in the manuscript.

– Figure 7: caption states that data is shown as mean+/- SEM though error bars are not shown. Panel B, scale at the bottom right corner is difficult to visualize. Maybe extend from 100um to 1000um with tick marks every 100um.

The figure has been edited and the scale bar increased to 500um proportional to the image size.

– With regards to figure 7 (and falling back on the fact that exosomes may contain similar or dissimilar content between IPF and controls)- can a comment be made on why control exosome skin does not heal? Though the difference in % healing is significantly greater than IPF exosomes, there is still a substantial impairment in wound healing.

We recognize that wounds treated with the control exosomes did not achieve complete healing. This is not surprising considering the age of the control exosomes. We have a manuscript prepared that will address age of exosomes and healing.

– Line 502: measured is misspelled.

Corrected

– U-IPFexo micro-RNA expression/MMP9 activity paragraph- Was the method for micro-RNA extraction from tissue described in the methods/supplement?

The methods section (Real Time PCR) has been updated with the PCR machine information.

– Line 507: Why would cell-specific exosomal micro-RNA content be less heterogenous than circulating/urine (excreted) exosomal micro-RNA content? Given that urine exosomes are unlikely to come from a single source would there not be more heterogeneity in urine?

The authors are not trying to suggest that the cargo of urine vs fibroblast exosomes are more or less heterogenous than each other. It is likely that there is heterogeneity in urine derived exosomes as well. However, the fibroblasts utilized in this investigation may also reflect heterogeneity in populations that have been isolated from populations of cells even within a fibroblastic foci as discussed by Tsuki et al.

– Tables 4 and 5 are difficult to interpret because significance symbols and text blend together.

The tables have been edited and the authors thank the reviewer for pointing this out.

– Is the corresponding experiment for Table 4 in mouse lung, human lung, both? Clarify.

Table 4 data are derived from mouse lung punches only. This has been clarified in the title of the table.

– Line 511-514: Conclusion regarding miR-34 does not seem to be consistent with what is shown in Table 4. Table 4 demonstrates a relative upregulation after urine exo versus a relative and seemingly significant downregulation with myofibroblast exo (though this is labeled as "NS"). Similarly, conclusions relating to miR-142 seem to be discrepant and seemingly significant based on the table data. Would recommend that MWU -tests for this data be revised. Can these discordant relationships be commented on?

The authors agree that we would expect the results from mouse lung punches to be similar regardless of the origin of the-exosomes. We have reanalyzed the data using MWU and found that the data for miR-142 is not significant. There was an input error for miR-34a, and the table has been edited as such, however the data is not significant. The reason for the difference between urine and fibroblast derived exosome outcomes is unclear and will require analysis with more data points. We are preparing to perform more experiments with human lung to determine differences.

– Line 532: Needs a period between the closing parenthesis and Bleo to start the sentence.

Thank you for pointing this out.

– Bleomycin + Control exosomes seem to have a protective effect against fibrosis, is this because of the different cell lines (myofibroblasts vs fibroblasts- with fibroblasts providing regenerative properties to the tissue?)

The authors acknowledge that there may be some “beneficial” effect of those exosomes, and it is tempting to suggest some regenerative properties. These and other important mechanisms are currently being explored in our laboratory.

– Line 529- in terms of body weight what can be said about non-cf bronchiectasis mice losing at least as much body weight as IPFexo mice (again in reference to are the non-cf bronchiectasis mice exosomes carrying "inflammatory" cargo).

It is likely that non-cf bronchiectasis derived exosomes carry an inflammatory cargo as they are continuously infected. This is an important aspect of the exosome cargo that the authors believe varies between different lung pathologies. Our ongoing studies are examining this question.

– Can a comment be made on the discrepancy between miR-142 in the punch biopsy experiment and the in vivo experiments? In the punch experiment miR-142 remains at stable levels whereas in the in vivo experiment it is decreased.

The punch experiments while illustrative were limited in number and the contribution of other factors including the immune system. It is likely that there are other factors important for changes in microRNAs that are beyond the aegis of this manuscript.

– Table 5. If ANOVA was performed a post hoc analyses (Tukey) would likely be more suitable.

ANOVA on this data has been completed and included in the revised manuscript.

– For the in vivo experiment: are these data concordant with myofibroblast/fibroblast derived exosome administration? It seems that this would be a relevant experiment to show that there is concordance between urine and cell-specific profibrotic changes.

The limited amount of fibroblast derived exosomes prevented those experiments from being performed.

– Discussion will likely need revision with changes. In line 704- again there is a definitive statement which should be avoided given small N.

The authors have edited the discussion and thank the reviewer for the suggestions.